# Presence of complete murine viral genome sequences in patient-derived xenografts

Zihao Yuan[1,2], Xuejun Fan[2], Jay-Jiguang Zhu [3], Tong-Ming Fu [2], Jiaqian Wu[3], Hua Xu[1], Ningyan Zhang [2], Zhiqiang An [2✉] & W. Jim Zheng [1✉]

Patient-derived xenografts are crucial for drug development but their use is challenged by issues such as murine viral infection. We evaluate the scope of viral infection and its impact on patient-derived xenografts by taking an unbiased data-driven approach to analyze unmapped RNA-Seq reads from 184 experiments. We find and experimentally validate the extensive presence of murine viral sequence reads covering entire viral genomes in patient-derived xenografts. The existence of viral sequences inside tumor cells is further confirmed by single cell sequencing data. Extensive chimeric reads containing both viral and human sequences are also observed. Furthermore, we find significantly changed expression levels of many cancer-, immune-, and drug metabolism-related genes in samples with high virus load. Our analyses indicate a need to carefully evaluate the impact of viral infection on patient-derived xenografts for drug development. They also point to a need for attention to quality control of patient-derived xenograft experiments.

[1] School of Biomedical Informatics, University of Texas Health Science Center at Houston, Houston TX, USA. [2] Texas Therapeutics Institute, Institute of Molecular Medicine, McGovern Medical SchoolUniversity of Texas Health Science Center at Houston, Houston, TX, USA. [3] Department of Neurosurgery, McGovern Medical School, University of Texas Health Science Center at Houston, Houston, TX, USA. ✉email: Zhiqiang.An@uth.tmc.edu; Wenjin.j.zheng@uth.tmc.edu

Preclinical evaluation of the effects of potential treatments using in vivo and in vitro platforms is essential for developing successful cancer treatments. As an important in vivo platform, Patient-derived xenograft (PDX) models are developed by implanting human tumor tissues in immune-deficient mice and have been considered a more faithful representation of the in vivo microenvironment for tumor growth[1–3] than cell culture. As of July 2020, over 4031 PDX models were deposited in the PDX finder[4], and at least 19,242 publications related to mouse models of cancer are deposited in the Mouse Tumor Biology Database[5]. The PDX market will be a $167.6 million enterprise by 2022[6], and there are 210 ongoing NIH-funded projects relevant to PDX, with a combined annual fiscal year budget of over $116 million (Supplementary Table 1).

Despite their critical role in cancer drug development, PDX models also face some significant challenges[7]. Viral infection was reported by two studies that isolated murine endogenous retrovirus from colorectal cancer PDX tumors[8] and PDX-derived cell lines[9]. A third study also detected murine viruses, such as xenotropic murine leukemia virus, in PDX tumors[10]. Even thought it is known that murine viruses can infect human PDX[11], viral infection was only investigated in these original studies with very small sample sizes (17–46 samples) and very limited assessment of its impact on PDX tumors; only one microarray study evaluated gene expression changes in a PDX-derived cell line rather than in PDX tumors[10]. Many questions remain about the extent of viral infections and how they impact PDX systems at the genome scale.

Next-generation sequencing (NGS) provides an unprecedented opportunity to evaluate viral infection of PDX tumors and their impact on the PDX genome. RNA-seq data generated from PDX tumors capture expressed RNA from PDX tumors and viruses that could infect murine stromal cells and PDX tumor cells. Unfortunately, in a typical RNA-Seq data analysis, sequencing reads are mapped to the human genome to evaluate expression levels of human genes or genome DNA variations. Unmapped reads—the "dark matter" of the sequencing data that most likely include mouse stromal cells and viral sequence reads—are disregarded. The value of analyzing unmapped reads is reflected in some recent projects assessing the landscape of viral associations in human cancer, such as the PCAWG project[12] and TCGA datasets[13]. In addition, tools have been developed to analyze these unmapped reads[14,15]. By analyzing these discarded reads together with human sequences, we can quantify the levels of gene expression for PDX tumors, mouse stromal cells, and murine viruses to thoroughly assess murine viral infection in PDX and its impact on the transcription profile of PDX tumors. Furthermore, RNA-Seq data can be used to detect viral sequences integrated into the PDX tumor genome, if any, to evaluate whether the integrity of PDX genome is compromised by viral infection.

In this work, we compare the "dark matter" of RNA-Seq data from 184 datasets generated from PDX tumors and single tumor cells to corresponding primary tumors and cells directly obtained from patients with no exposure to mice or murine viruses. Our data-driven approach[16] assesses the sequences with non-human origin and characterizes the landscape of murine viruses in PDX. We find extensive presence of the complete murine viral genome in 170 datasets evaluated, and many cancer-, immune-, and drug metabolism-related genes have significantly changes in expression levels in high virus load samples.

## Results

**Presence of murine viruses in PDX models**. We evaluated the presence of viral sequences in PDX tumors (Table 1) starting from publicly available RNA-seq raw reads (excluding single-cell

**Table 1 Sample origins of sequenced PDX tumors and non-PDX controls.**

| Sample type | | PDX tumor | Primary tumor | Cell line |
|---|---|---|---|---|
| Breast cancer | PDX | ✓ | | |
| | Control | | | ✓ |
| Glioblastoma | PDX | ✓ | | |
| | Control | | ✓ | ✓ |
| Lung cancer | PDX | ✓ | | |
| | Control | | ✓ | ✓ |
| Ovarian cancer | PDX | ✓ | | |
| | Control | | | ✓ |
| Bladder cancer | PDX | ✓ | | |
| | Control | | ✓ | |
| Colorectal cancer | PDX | ✓ | | |
| | Control | | ✓ | ✓ |
| Pancreatic cancer | PDX | ✓ | | |
| | Control | | ✓ | |

Controls are samples not exposed to mouse environment and murine viruses, and are either corresponding primary tumor, primary cell culture, or both.

sequencing; see Supplementary Table 2a, 2b, 2c) and analyzed sequence reads that cannot be mapped to the genome of PDX tumors. The data we used are carefully selected from the controls of these experiments, and most samples were from tumors directly obtained from patients and then grafted into mice with no modification. Therefore, these samples were not manipulated, engineered, or luciferized, thereby allowing us to avoid any artificial viral sequences that could complicate our analyses.

We started with data generated by RNA-Seq, a NGS technique used to sequence and quantify existing RNA molecules from cells. We found extensive presence of murine viral RNA sequences in human-derived PDX tumors (170 of 184 conventional RNA-seq samples) compared with tumors cultured in a xenograft-free environment (Fig. 1a, Table 1, Table 2, Supplementary Table 3, Supplementary Data 1). The murine viruses span most of the PDX tumor virome—all unmapped reads that can be mapped to any viral genome, regardless of their host origin. This pattern was observed in PDX models of bladder cancer (Wilcoxon-rank test, $p$-value $= 6.91E-08$); breast cancer (Wilcoxon-rank test, $p$-value $< 2.2E-16$); colorectal cancer (Wilcoxon-rank test, $p$-value $= 1.55E-03$); glioblastoma (Wilcoxon-rank test, $p$-value $= 1.29E-04$); lung cancer (Wilcoxon-rank test, $p$-value $< 2.2E-16$); ovarian cancer (Wilcoxon-rank test, $p$-value $= 9.824E-07$); and pancreatic cancer (Wilcoxon-rank test, $p$-value $= 1.192E-10$). We also observed relatively high levels of murine virus in NSG mice compared to wild-type mice (Wilcoxon-rank test, $p$-value $= 1.72 E-04$). This was an expected outcome given that NSG mice are immunocompromised.

Murine viruses also made up a higher proportion of total sequenced reads in PDX tumor samples compared with corresponding primary tumors obtained directly from patients (Supplementary Fig. 1, 2, Supplementary Table 4, column IX–XII) with bladder cancer (Wilcoxon-rank test, $p$-value $= 6.91E-08$); breast cancer (Wilcoxon-rank test, $p$-value $< 2.2E-16$); colorectal cancer (Wilcoxon-rank test, $p$-value $= 1.55E-03$); glioblastoma (Wilcoxon-rank test, $p$-value $= 1.29E-04$); lung cancer (Wilcoxon-rank test, $p$-value $< 2.2E-16$); ovarian cancer (Wilcoxon-rank test, $p$-value $= 3.93E-06$); and pancreatic cancer (Wilcoxon-rank test, $p$-value $= 1.192E-10$). PDX tumors had even higher levels of murine viral loads than NSG and wild-type mice—a finding consistent with the previously reported lack of robust anti-infection mechanisms in tumor cells[17,18].

Figure 1b shows that nearly half of the viruses in human origin PDX tumors are murine leukemia virus, and another 22% from the provirus for various endogenous murine retroviruses. Besides

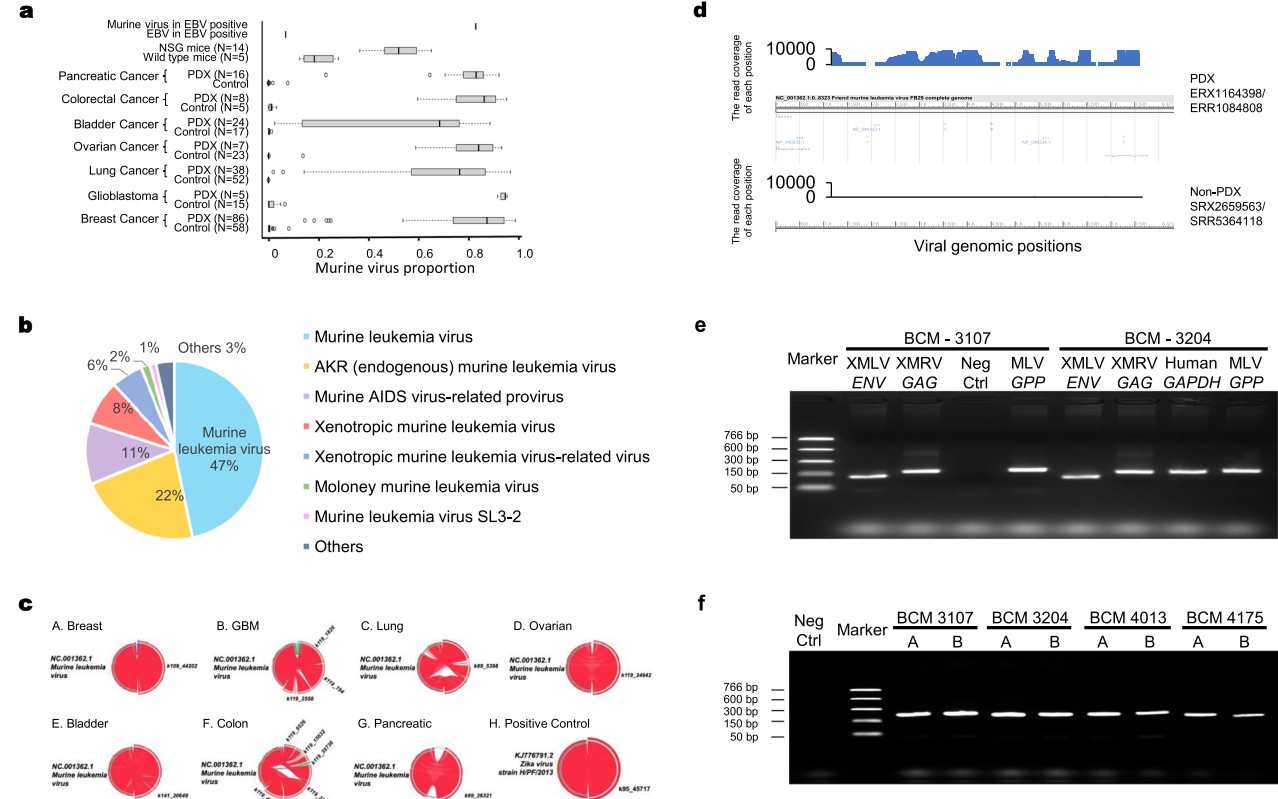

**Fig. 1 Abundant murine virus sequence reads found in RNA-Seq data generated from PDX tumor cells. a** The proportion of murine virus reads in the total virome in various PDX tumor samples and corresponding controls, wild-type mice and NSG mice. Box limits: 25th and 75th percentiles; center line: median; whiskers: 1.5x interquartile range from box limits; dots: outliers. **b** Major categories of murine virus as identified by sequence reads from all PDX cancer samples. **c** Assembly of full viral genome from RNA-Seq reads. A-G: murine leukemia virus from seven types of PDX. H. Zika-virus-infected human brain organoids as positive control. **d** Sequence reads from breast cancer PDX (top) but not from non-PDX samples (bottom) cover the entire murine leukemia virus genome. **e** Amplification of XMLV *ENV*, XMRV *GAG*, and MLV *GPP* gene fragments from genomic DNA preparations of two breast cancer PDX tumor samples. The human *GAPDH* gene was used as a control to validate the experimental system. Reaction without template is considered as negative control. **f** Amplification of the MLV *GPP* gene fragment from 4 breast cancer PDX models with duplicates (A, B). PCR amplicfication was performed twice for each experiment. Source data for (**a**, **b**, **e**, and **f**) are provided in Source Data File.

these previously described viruses[8,10,19], we also identified others, such as AIDS virus-related provirus or AKT8 retrovirus, not formerly reported in PDX tumors (Fig. 1b). Most of these viruses were present in different PDX tumors, indicating that the mouse environment and the general lack of robust anti-infection mechanism in tumor cells[17] may play important roles. Figure 1c (A-G) shows that the murine leukemia virus is fully detected in all seven types of PDX tumors but not in corresponding primary tumors or cell lines (Table 2, PDX v.s. CON). The analysis method is validated by the positive control detecting full Zika-virus sequence using RNA-Seq data from Zika-virus-infected organoids (Fig. 1c, h). Furthermore, the detected viral reads cover the entire viral genome (Fig. 1d), indicating a functional viral genome with full capacity for infection.

We performed PCR experiments to confirm the extensive presence of murine viruses in PDX tumors. High-quality genomic DNA preparations extracted from four different breast cancer PDX tissues were obtained from the Baylor College of Medicine PDX Core. PCR primers were designed according to previously published work to allow us to detect various murine virus-specific genomic regions[9,20–22]. These virus-specific primers do not match any human genome regions, any human retrovirus sequences, or any artificial sequences such as vectors used for GFP or luciferase markers as confirmed by the in-silico PCR tool provided by the UCSC Genome Browser. In addition, these primers target viral-specific regions in xenotropic murine leukemia virus (XMLV), a virus known to infect human cells but not mouse cells. We first used two PDX DNA samples (BCM-3107 and BCM-3204) and detected a single fragment of the *ENV* gene of XMLV, the *GAG* gene of XMRV and the *GPP* gene of MLV, respectively, with expected sizes (Fig. 1e). In addition, we used all four samples to detect the MLV *GPP* gene segment (Fig. 1f).

These murine viruses are retroviruses and can only be in DNA form while they are in an active infectious state. The fact that we can detect targeted viral DNA fragments from all tested PDX tumor samples indicates that: (1) detected fragments are not from free viruses from mouse blood or tissue contamination, since they only have genomic RNAs and cannot be amplified by the polymerase we used in PCR experiments; and (2) viruses exist extensively in an active infectious state in the intermediate replication DNA form in all the PDX tissues tested, because these retroviruses only produce DNA fragments during their replication phase. Our experiments confirmed our computational analysis that these viral sequences are present extensively in PDX tumor samples (Fig. 1f) and are not an artifact of any kind. Our observation raises some significant questions: 1) Where do the viruses come from? 2) How extensive is the presence of these viruses? 3) Are there any biological implications and consequences associated with the presence of these viruses?

**Table 2 Average reads of murine viruses from PDX and non-PDX cancer samples (CTR) with total reads count over 200.**

| Viral Names | Breast cancer PDX | Breast cancer CTR | GBM PDX | GBM CTR | Lung cancer PDX | Lung cancer CTR | Ovarian cancer PDX | Ovarian cancer CTR | Bladder cancer PDX | Bladder cancer CTR | Colorectal cancer PDX | Colorectal cancer CTR | Pancreatic cancer PDX | Pancreatic cancer CTR | Wild-type mouse | NSG mouse |
|---|---|---|---|---|---|---|---|---|---|---|---|---|---|---|---|---|
| Abelson murine leukemia virus | 265 | 0 | 126 | 271 | 177 | 0 | 1131 | 485 | 233 | 0 | 651 | 31 | 4861 | 0 | 1477 | 136 |
| AKR (endogenous) murine leukemia virus | 73286 | 0 | 234905 | 213 | 70546 | 0 | 414658 | 0 | 87206 | 40 | 229581 | 0 | 22206 | 0 | 671 | 1824 |
| AKT8 retrovirus | 2051 | 242 | 3305 | 517 | 1672 | 1 | 15855 | 539 | 1076 | 0 | 6047 | 17 | 16499 | 82 | 4262 | 352 |
| Curionopolis virus | 29 | 0 | 0 | 0 | 25 | 0 | 156 | 0 | 9 | 0 | 42 | 0 | 309 | 0 | 240 | 20 |
| Endogenous mouse mammary tumor virus Mtv1 | 68 | 0 | 0 | 0 | 57 | 0 | 692 | 0 | 43 | 0 | 36 | 0 | 1815 | 0 | 126 | 184 |
| Exogenous mouse mammary tumor virus | 74 | 0 | 0 | 0 | 58 | 0 | 698 | 0 | 49 | 0 | 24 | 0 | 2146 | 0 | 157 | 247 |
| Friend spleen focus-forming virus | 121 | 0 | 265 | 15 | 1082 | 0 | 839 | 0 | 44 | 0 | 1673 | 0 | 2971 | 0 | 567 | 88 |
| Harvey murine sarcoma virus | 9 | 2 | 1 | 2 | 5 | 0 | 40 | 8 | 7 | 0 | 35 | 0 | 247 | 1 | 266 | 38 |
| HoMuLV murine leukemia virus | 65 | 0 | 0 | 1 | 104 | 0 | 36 | 0 | 15 | 0 | 15 | 0 | 5378 | 0 | 172 | 17 |
| Intracisternal A-type particle IAP | 57 | 0 | 6 | 1 | 63 | 0 | 453 | 1 | 52 | 0 | 160 | 0 | 1042 | 0 | 2874 | 103 |
| Lactate dehydrogenase-elevating virus | 1066 | 0 | 0 | 0 | 1069 | 0 | 0 | 0 | 0 | 0 | 0 | 0 | 0 | 0 | 0 | 0 |
| MLV-related virus CFS | 514 | 0 | 4 | 10 | 1071 | 0 | 1274 | 617 | 910 | 0 | 1485 | 0 | 33164 | 0 | 2216 | 226 |
| Moloney murine leukemia virus | 2575 | 0 | 11735 | 66 | 4747 | 0 | 20787 | 177 | 5780 | 0 | 9366 | 0 | 28222 | 0 | 1867 | 193 |
| Moloney murine sarcoma virus | 2910 | 13 | 2149 | 26 | 1829 | 0 | 19020 | 0 | 2158 | 0 | 8932 | 0 | 12031 | 1 | 3599 | 361 |
| Mouse mammary tumor virus | 122 | 0 | 0 | 0 | 123 | 0 | 1502 | 0 | 68 | 0 | 81 | 0 | 2067 | 0 | 324 | 628 |
| Murine AIDS virus-related provirus | 10788 | 0 | 34188 | 69 | 5913 | 0 | 39649 | 1098 | 6006 | 0 | 14073 | 0 | 451126 | 0 | 10167 | 421 |
| Murine leukemia virus | 124602 | 0 | 360905 | 628 | 123725 | 0 | 721918 | 0 | 113269 | 0 | 377134 | 0 | 566305 | 0 | 27359 | 4756 |
| Murine leukemia virus SL3-2 | 3520 | 0 | 4907 | 54 | 9808 | 0 | 18535 | 0 | 1318 | 0 | 12369 | 0 | 8243 | 0 | 1339 | 326 |
| Xenotropic murine leukemia virus | 18111 | 0 | 78568 | 69 | 10516 | 0 | 170003 | 0 | 13798 | 0 | 67339 | 0 | 62027 | 0 | 3887 | 370 |
| XMRV | 7221 | 0 | 20383 | 26 | 1992 | 0 | 116378 | 0 | 53253 | 0 | 43398 | 0 | 55809 | 0 | 3095 | 460 |

**No correlation between the number of sequence reads from murine viruses and from mouse stromal cells contaminating PDX tumor.** A phylogenetic analysis of the *ENV* and *GAG* genes shows that these murine viruses exhibit distinct differences from human endogenous retrovirus (Fig. 2a, b), indicating that our observed viral sequence reads are not from human endogenous retrovirus.

Another possible source for detected viral sequences may be PDX tumor samples contaminated with mouse stromal cells, since PDX tumors are implanted into host mice. If this is the case, there could be a significant correlation between the number of mouse cells and the number of viral sequence signals we observed. We evaluated this possibility by first analyzing the presence of sequence reads that can be mapped to the mouse genome—an indication of mouse stromal cell contamination. The observed mouse sequence reads (Supplementary Table 4, column V) indicate that PDX tumors are indeed contaminated with mouse stromal cells.

As we mapped all the sequence reads to the human genome to generate unmapped reads, some mouse reads from the contaminated tissue could have been mapped to the human genome due to sequence homology. To accurately evaluate the extent to which mouse tissues are present in PDX, we used NGS data from 14 mouse control samples and mapped them to the human genome first. The unmapped reads were then mapped to the mouse genome. Ratios between mouse and human genome mapped reads were calculated (Supplementary Table 5) and used to infer the number of true mouse reads in the PDX sequence reads not mapped to the human genome (Supplementary Table 4, column VI). The adjusted mouse reads were used to calculate the percentage of the mouse reads in the total sequence reads (Supplementary Table 4, column VIII).

We quantified the presence of murine viruses in PDX by mapping the unmapped reads from PDX samples to the genomes of all viruses (Supplementary Table 4, column IX), and identified reads belonging to murine viruses (Supplementary Table 4, column XI). The percentage of murine virus reads in all viral reads was then calculated (Supplementary Table 4, column XII).

Mouse reads (Supplementary Table 4, column VI) were more common than murine virus reads by reads count (Supplementary Table 4, column IX). However, since the mouse genome is much larger than the viral genome, the overall read depth for mouse is much less than that for murine viruses (Fig. 2c, d). Therefore, we normalized reads by the size of the source genome for a more accurate comparison. After this adjustment, the viral genome reads (Supplementary Table 4, column X) were much higher than mouse reads (Supplementary Table 4, column VII), indicating low levels of contamination by murine tissues compared to viral gene expression.

We then investigated whether there is a significant correlation between the number of reads from the mouse genome and murine viruses. We plotted the percent of mouse reads in total sequences (Supplementary Table 4, column VIII) against the percent of murine virus reads in the PDX virome (Supplementary Table 4, column XII). There were no correlations between the numbers of these mouse-specific reads and the proportion of the murine virus in the PDX virome (Pearson correlation test: 0.03, *p*-value = 0.6847) (Fig. 2e), an indication that murine viruses may be produced by not only contaminated mouse stromal cells, but also PDX tumor cells.

Besides murine viruses, we also identified human viruses long associated with PDX tumors or human cancer in general. In previous reports, in some cases, PDX tumors in the mouse host were replaced by human lymphocytes transformed by the Epstein–Barr virus (EBV)[12,23,24]. EBV sequence reads were present in some PDX tumor samples we tested, such as ERR1084820 and ERR1084816. In these samples, the EBV reads covered the whole genomic region (see Supplementary Fig. 3). On

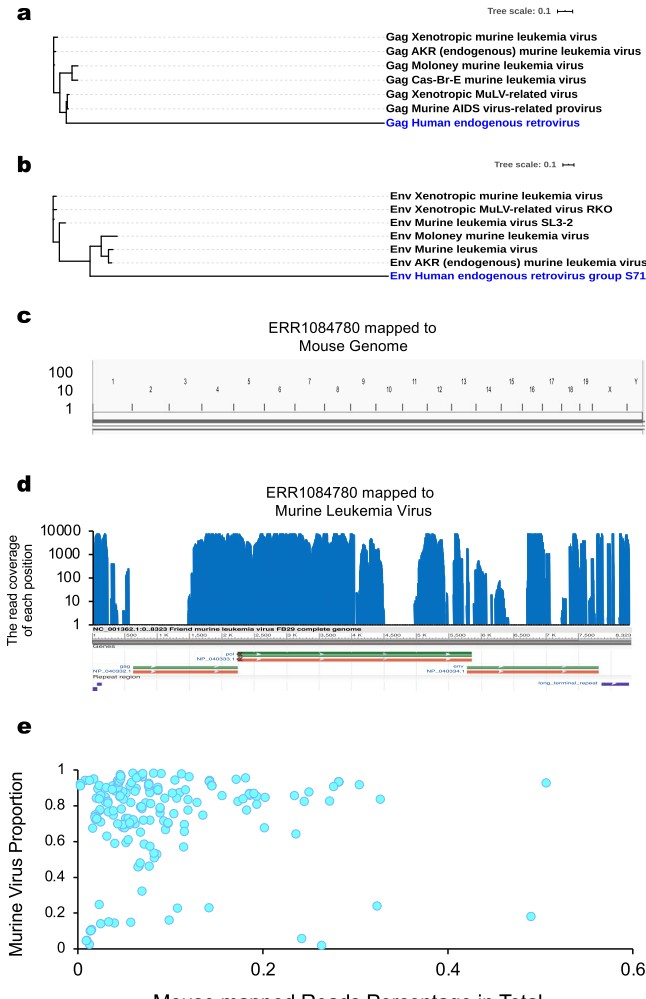

**Fig. 2 Murine viral sequences from PDX are not from human viruses or correlated with mouse tissue contamination. a** Phylogenetic analysis of *GAG* genes of murine (black) and human viruses (blue). **b** same analysis performed for *ENV* genes. **c** Low level of reads from PDX that can be mapped to mouse reference genome. **d** High-level murine virus reads mapped to the MLV viral genome. **e** No significant correlations were found between the number of mouse-specific (*x*-axis) and murine viruses reads (*y*-axis) across all PDX samples examined ($n = 184$). Source data for **e** are provided in Source Data File.

the other hand, although some have reported associations between human cytomegalovirus (HCMV) and human cancer[25], especially glioblastoma multiforme[26], we did not detect HCMV sequences in the human tumors we analyzed.

**Single-cell sequencing indicates the presence of intracellular viruses.** We evaluated the presence of murine viruses inside cells from PDX tumors using data generated from single-cell sequencing experiments where RNA molecules from a single cell were isolated and sequenced. Although single-cell sequencing sometimes can be criticized as low throughput[27] and fewer raw reads can be mapped to the reference genome compared with conventional RNA-seq results (Wilcoxon-rank test, *p*-value < 2.2e−16 and Supplementary Table 6, using the housekeeping gene *ACTB* in lung cancer as an example), it provides direct evidence of RNA molecules inside a single cell.

We found a significantly higher number of murine viral sequence reads in single-cell-sequenced PDX lung cancer samples than in corresponding primary tumors (Wilcoxon-rank test, *p*-values = 1.99E−06) (Fig. 3a) with no mouse exposure. In addition, the viral reads from PDX samples cover the entire genomic region of the viruses, despite the low read count due to limitations of the single-cell sequencing technology (Fig. 3b, c). In control samples, no viral reads could be mapped to the murine viral genome (Fig. 3d). Furthermore, there were no significant differences between PDX and primary cell cultures for the small number of reads that could be mapped to the mouse genome (Fig. 3e). These reads are scattered across the entire mouse genome at low depths of coverage; therefore, they are not real mouse reads but sequence noise. Unlike bulk sequencing performed on PDX tumors or cell cultures, single-cell sequencing detects RNA molecules from individual cells and the protocol makes it impossible to contaminate free viruses. Since most sequence reads from these cells were mapped to the human genome, we are certain the sequences are from individual human tumor cells. These results provide direct evidence to support our hypothesis that murine-origin viruses are not merely contamination from murine tissues or blood, but are inside PDX tumor cells.

**Presence of chimeric sequence reads with both murine viral and human sequences.** Besides extensive viral sequence reads, we also found chimeric sequence reads that contain both murine viral genome and human sequences from RNA-seq data generated from PDX tumors (Fig. 4). Since some murine retroviruses can integrate their genome into their host genome, transcription from the integration sites could generate these types of chimeric sequences[28]. We identified chimeric RNA-seq reads from unrelated studies of PDX tumors at depth of coverages ranging from 10x to 242x, containing both different parts of murine virus such as LTR regions, and human genome sequences from different chromosomes (Fig. 4, Supplementary Data 2). Such depth of coverage indicates that these chimeric reads are not random sequencing errors, but originate from viral integration sites as reported before[28].

**Implications of the extensive presence of murine viruses for PDX systems.** The extensive presence of murine viral sequences in PDX samples and the evidence supporting the existence of viruses inside PDX tumor cells raise the question whether these viruses could be associated with any changes in PDX tumors. We found significant differences in gene expression between samples with low and high murine virus loads. Fig. 5a shows that among the top 200 most differentially expressed genes, many genes involved in apoptosis, cell death, or tumor necrosis were down-regulated in groups with high murine virus loads (Fig. 5a, Supplementary Fig. 4a, b). These genes were also enriched with ontology terms related to immune regulation and important signaling and disease pathways (Supplementary Fig. 5a–c). In addition, in differentially expressed genes among groups with high and low virus loads in bladder cancer PDX treated with the PI3K inhibitor pictilisib, pathways related to drug metabolism were significantly enriched (Fig. 5b).

The association between virus load and tumor gene expression change was further investigated. It is possible that diverse gene expression profiles may exist in primary tumors from different patients, leading to different levels of viral growth in PDX tumors, as we observed above. On the other hand, primary tumors may have similar gene expression profiles, but these profiles may be markedly impacted by the exposure to murine viruses after tumors are implanted in mice. Both scenarios could result in the same observed association between viral load and gene expression variations. To evaluate these possibilities,

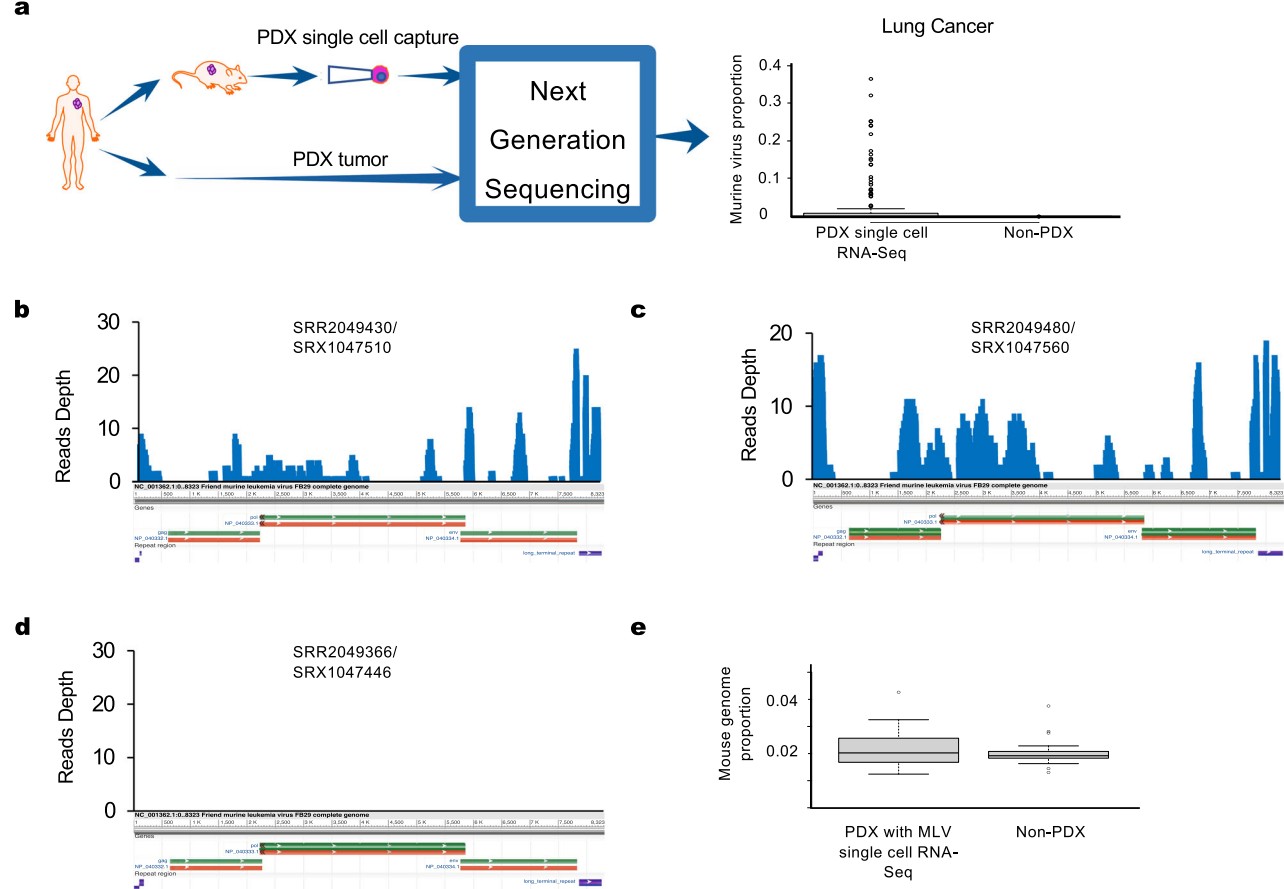

**Fig. 3 Single-cell sequencing support the presence of murine viruses inside PDX tumor cells. a** Relative proportions of murine viruses in the virome of lung cancer PDX cells are significantly higher than in the cells from the primary lung cancer samples as measured by single-cell RNA-Seq (PDX: $n = 1147$; non-PDX: $n = 152$). **b, c** Viral reads from single-cell sequencing of PDX-derived tumor cells cover the entire genomic region of MLV. **d** No single-cell sequencing reads from non-PDX control samples were observed or mapped to the viral genome. **e** The single-cell sequencing reads that can be mapped to the mouse reference genome are very low in both PDX-derived tumor cells and non-PDX control cells (PDX: $n=9$; non-PDX: $n = 152$). For box plots in (**a**) and (**e**), Box limits: 25th and 75th percentiles; center line: median; whiskers: 1.5x interquartile range from box limits; dots: outliers. Source data for **a** and **e** are provided in Source Data File.

we performed a principal component analysis on all PDX samples we used and the corresponding primary tumor controls. All primary tumors formed a tight cluster, while PDX samples were scattered at a significant distance from each other (Fig. 5c). This analysis indicates that primary tumors have very similar gene expression profiles before being transplanted into mice for PDX formation and are unlikely to contribute to different levels of virus load in PDX tumors. On the contrary, it is more likely that the observed variation of gene expression occurred after primary tumors were transplanted and exposed to the viruses.

Since the virus load-associated changes in gene expression could affect drug response in PDX models, we investigated whether viral infection of human cancer cells can mimic gene expression changes in PDX tumors with high murine virus load. Similar gene expression patterns emerged between PDX tumors with high virus load and previously reported human cell lines infected by oncolytic virus (Fig. 5d). For example, genes such as *IRF5* and *STAT1* are downregulated in lung cancer cells after infection by viral RNA[29], similar to PDX tumor groups with high murine virus load in our analyses. The *TERT* gene is upregulated among the PDX tumor groups with high murine virus load and in adenovirus-treated human cancer cell lines[30].

## Discussion

Genetic instability of cell lines[31–33] and cell line contamination[34–37] reveal widespread and systematic issues that could significantly impair the reliability and reproducibility of scientific data. Likewise, murine retroviruses such as murine leukemia virus or EBV[38] in PDX models have been reported in small datasets[8–10]. However, these assessments were very limited due to lack of available data. NGS technology captures a wide variety of data ranging from RNA-seq to single-cell sequencing. By using these large numbers of data that were previously unavailable, we found extensive presence of murine viruses in PDX samples (Fig. 1a). These findings are experimentally confirmed as a biological reality, not informatics artifacts (Fig. 1e, f). We also identified many different types of viruses in PDX tumors not previously reported. Our work revealed extensive and persistent presence of murine viruses in PDX models not shown before and raises serious concerns about their impact on PDX models as a critical platform for cancer drug development.

By analyzing sequencing reads, we showed that there is mouse stromal cell contamination in PDX tumors, and it is likely some of the murine viruses we detected are from these cells. However, the lack of correlation between the number of mouse stromal cells and murine virus load, the extensive presence of chimeric sequence reads with high read depth containing both viral and

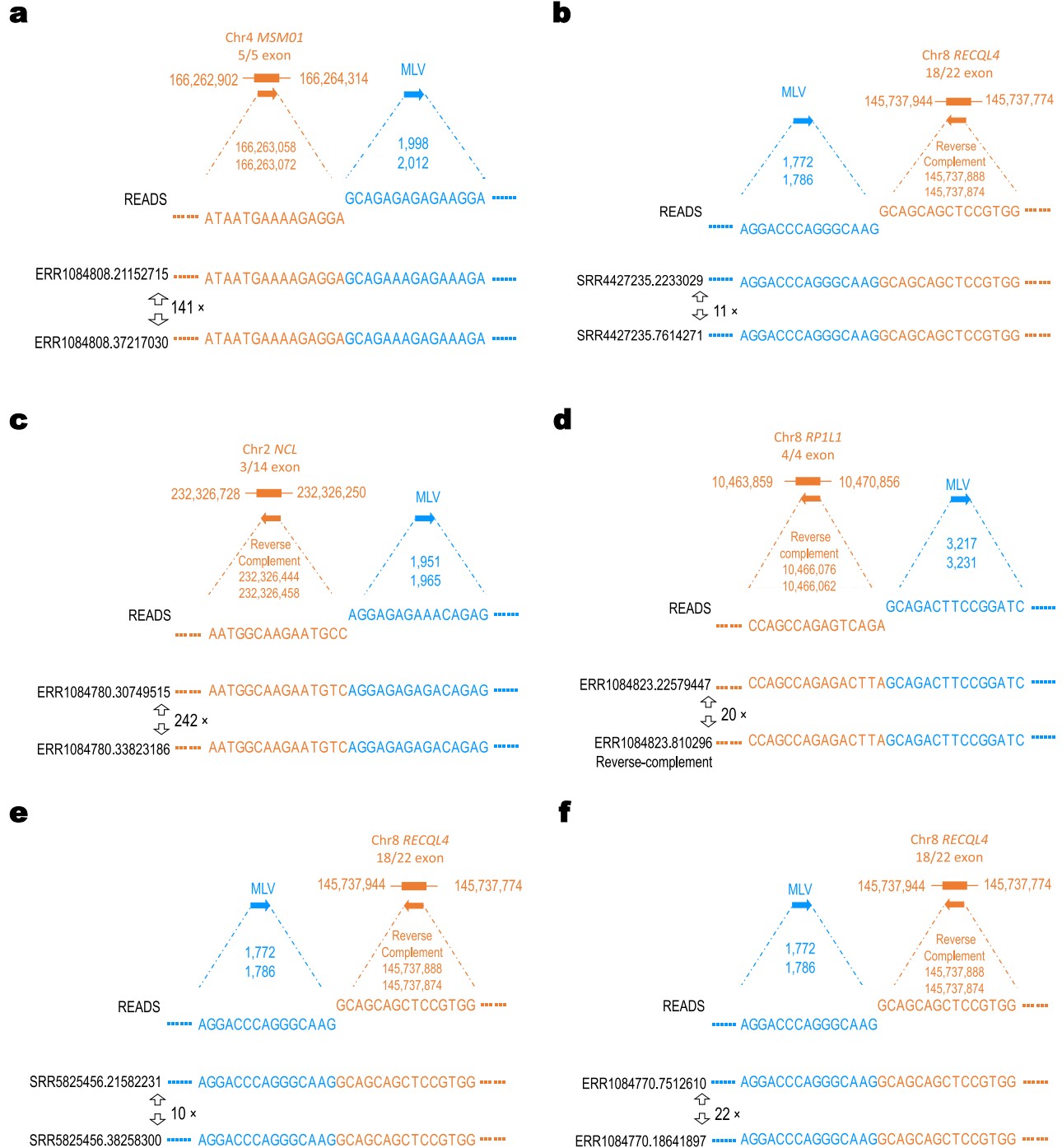

**Fig. 4 Chimeric sequence reads containing both viral (blue) and human (orange) genome sequences were identified from PDX RNA-Seq reads for six tumor types, indicating the integration of murine viral genome.** The chromosome, the gene name and the exon within which the integration site located is also provided. **a** Breast Cancer. Integration site is on chromosome 4, within the fifth exon of the *MSM01* gene which has five exons. **b** GBM. Integration site is on chromosome 8, within the 18th exon of the *RECQL4* gene which has 22 exons. **c** Lung Cancer. **d** Ovarian Cancer. **e** Bladder Cancer. **f** Colon Cancer. The genomic position and the orientation from human and virus genome are marked. The depth of coverage is labeled between reads. The detailed information is provided in Supplementary Data 2.

human genome sequences, and viral sequences in single-cell sequencing data from PDX tumor cells without any indication of mouse sequence contamination, all suggest that human tumor cells may also be infected by murine viruses and support viral replication persistently. As our bioinformatics approach relies on the availability of existing data from a wide variety of experimental conditions, the ultimate validation would be experimental

investigation to confirm the presence of the virus inside the cell in these conditions. Nevertheless, our observation is consistent with previous findings that xenotropic viruses can infect human tumor cells[11]. Our data-driven approach enabled us to assess the impact of viral infection at a genome scale from 184 PDX experiments, gaining a much deeper insight into the scope of murine viral infection in PDX than previously reported.

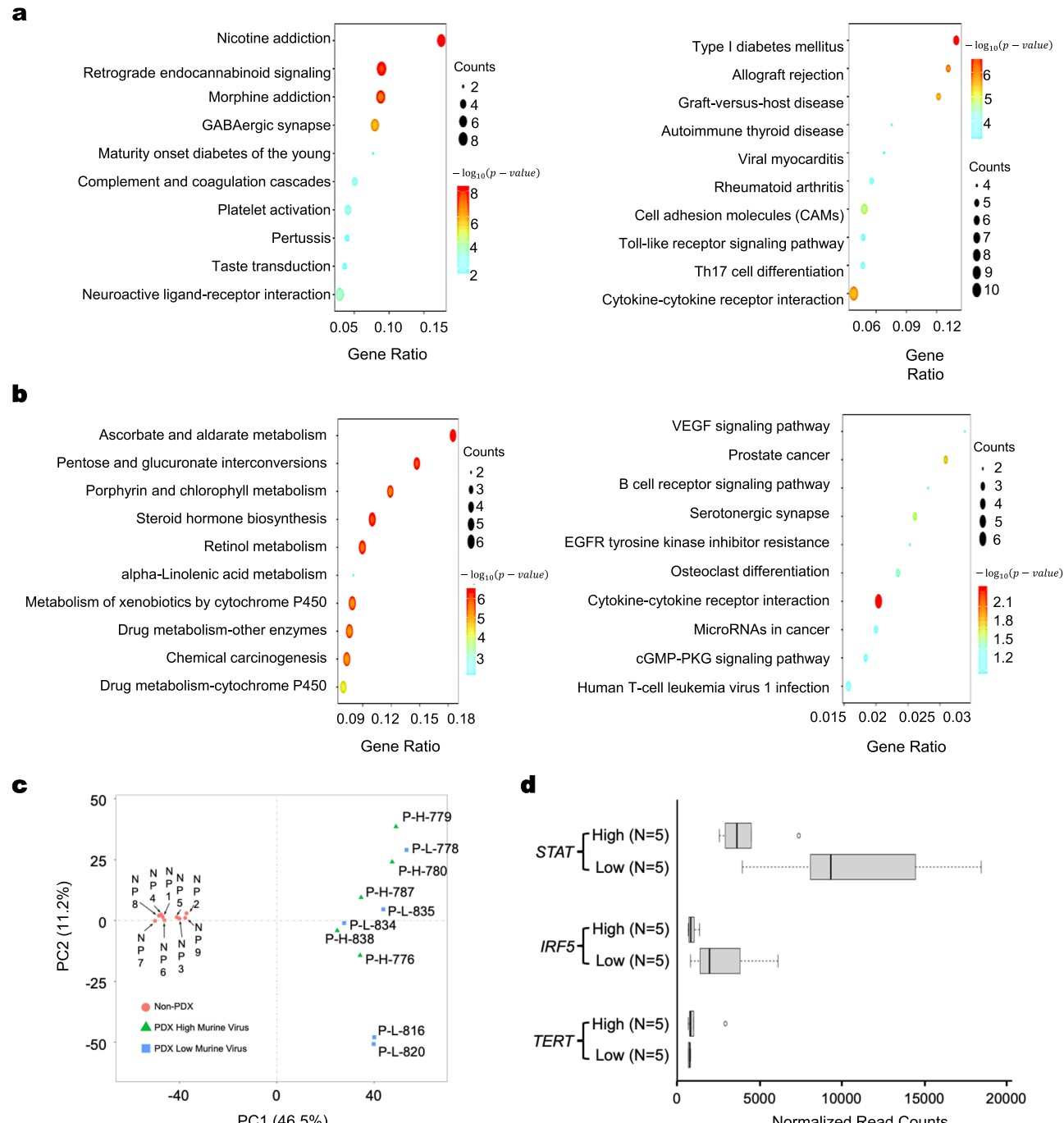

**Fig. 5 Impact of murine viral infection on gene expression in PDX tumors.** Transcription profiles of lung cancer PDX samples with highest and lowest viral numbers. Pathways and functions were enriched among: **a** the top 200 upregulated (left) and downregulated genes (right) in the 5 PDX samples with the most murine viral infection. **b** Transcription profiles of pictilisib-treated bladder cancer PDX samples with highest and lowest viral numbers. Pathways and functions were enriched among the top 200 upregulated (left) and downregulated (right) genes in the samples with the most murine viral infection. **c** Primary tumors before creation of PDX models have consistent gene expression profiles and form a major component in principal components analysis, compared to PDX tumors with viral infection and very diverse transcription profiles. **d** Three genes (*STAT*, *IRF5,* and *TERT*) differentially expressed between PDX tumors with highest (High, $n = 5$) and lowest (Low, $n = 5$) levels of murine viruses. There genes are also differentially expressed in oncolytic virus-infected lung cancer. Box limits: 25th and 75th percentiles; center line: median; whiskers: 1.5x interquantile range from box limits; dots: outliers. Source data for (**a**, **b**, and **d**) are provided in the Source Data File. Datasets used to create **c** are listed in Supplementary Table 2.

We observed strong correlations between murine virus loads and the expression of many important human genes or pathways that are crucial in cancer research and treatment. For example, in PDX tumors with high viral load, we observed downregulation of some immune genes, such as CD80, with essential roles in immune therapy deployment. If such downregulation occurs in lymphocytes within PDX tumors, it could suppress the expression of the *PD-L1* receptor and hence suppress restriction of T cell activation[39], leading to relatively high activity of T lymphocytes in viral-infected PDX models. The direct outcome would be a seemingly successful immune therapy that would not work on virus-free tumors in patients, if our observed gene expression change is indeed directly

related to murine viruses in the PDX environment. This concept is consistent with the observation that some PDL1 inhibitors, such as durvalumab, suppressed tumor growth in human tumor xenograft models[40] but failed in clinical trials[41]. Our findings indicate that how the extensive presence of murine virus may affect immune gene expression in PDX models warrants investigation, especially when developing drugs for immune therapy.

Our findings point to a possible way of recreating the high virus load tumor environment in cancer patients by oncolytic or other viruses, since we observed some shared gene expression changes in tumor cells after infection by these viruses and in PDX infected by the murine viruses. If such an environment causes tumor cells in a patient to have similar gene expression changes as we observed in PDX tumors, it might modify tumor sensitivity to drugs. This could in turn to help recover some drug leads that worked well in PDX models with high virus load, but not in clinical trials where the primary tumors of participants are free of viruses. In support of this idea, oncolytic viral infection can boost efficacy of drug leads. For example, H-1PVs, a rat protoparvovirus used as an oncolytic virus for clinical treatments, significantly improves the efficacy of glioblastoma treatment when combined with the anti-angiogenic drug bevacizumab or immune checkpoint blockade[42]. In addition, oncolytic viruses could stimulate the immunosuppressive microenvironment in tumors and attack tumor cells synergistically with immunotherapies[43].

PDX models have been and will remain an essential part of cancer research and drug development given their advantages over cell culture. However, the extensive and persistent presence of murine viruses and the strong association between virus load and gene expression changes cannot be ignored. Comprehensive and thorough evaluation of the impact of viral infection on PDX should occur to fully understand their effect on drug development. Given mature technologies such as CRISPR-Cas9[44], one solution could be to create a virus-free mouse strain for PDX studies.

Our results also indicate the need for more quality control in experiments using tissues of human origin in murine xenografts. For such experiments, conventional sanitation and pathogen control, as well as risk assessment of viral infection, should be in place. Surprisingly, the viruses we detected are not included in commercial solutions provided by companies such as IDEXX BioAnalytics who provides virus monitoring and detection products to most, if not all, PDX core facilities[45]. To prevent viral infection and transmission, animal models should be tested regularly and extensively for infections by as many viruses as possible. Such practices could have a profound impact on the success of cancer drug development on PDX models.

## Methods

**Data.** We used sequencing data from different sources, including (1) PDX tumors or PDX—patient-derived xenografts (sometimes called primagrafts), tumors of human origin grafted into and grown in mice; (2) primary tumors—tumor explants directly obtained from patients without any type of culture or treatment and were not exposed to mouse or murine viruses; (3) PDX cell lines—cell lines derived from the PDX tumor; (4) primary cell cultures—cell lines derived from patient tumor explants with no exposure to mouse or murine viruses; or (5) cell lines—lab cultured cell lines mostly available from the American Tissue Culture Collection. We use the above naming conventions throughout this work.

We searched the NCBI Sequence Read Archive (SRA)[46] and the literature to identify all NGS studies using PDX tumors. We limited the studies to samples sequenced on an Illumina platform. Samples, experiments, and their accession numbers are provided in Supplementary Table 2b. Raw reads for RNA-seq analyses were downloaded from the SRA database[46]. Before analysis, the raw reads were trimmed by Trimmomatic to remove adapter sequences, ambiguous nucleotides, and potential contamination[47].

All the datasets used in this study are free available from publicly resources without restriction. This study is solely a data re-analysis project with a simple molecular biology experiment for validation purpose without animal or human participants involvement, therefore is free from any ethical regulations or institutional protocol approval.

**Reads mapping and analysis.** The reads were first mapped to the human and mouse reference genomes using STAR[48] or Bowtie2[49]. Proportions of reads that could be mapped to the human and mouse reference genomes are detailed in Supplementary Tables 4 and 6.

Viral sequences were detected with the pathogen discovery program READSCAN[50]. READSCAN was conducted with default parameters. Reads were considered as a viral sequence if they had at least 10% coverage of the reference virus sequences. The viral genomes were assembled in Megahit in the sensitive model[51], and synteny with the viral genome was assessed with Circoletto[52]. The relative proportion of the murine virus load (or murine virus load) was calculated as the number of reads aligned to the murine viruses divided by the total number of viral reads mapped to any viral genome (see Supplementary Table 4).

Viruses of murine origin were identified through the Virus-Host DB[53], NCBI taxonomy databases, and previous literature (Supplementary Table 3). Viral integration sites were analyzed via Virus-Clip[54], and genomic sequences were viewed based on hg19 in UCSC Genome Browser[55]. Principal component analyses were conducted based on the average numbers of reads of murine virus in PDX tumors and corresponding primary tumor controls in Omicshare[56].

The RNA-seq reads were mapped to the MLV reference genome with Bowtie2[49], and the read depth was summarized via the depth function by the Samtools[57].

**Detection of murine viruses by PCR.** Supplementary Table 7 describes the PDX samples used for PCR analysis. The genomic DNA was isolated by the Baylor College of Medicine PDX Core Facility using a Qiagen DNeasy Blood and Tissue kit. DNA concentrations were measured using Nanodrop. We followed previously published work[9,20–22] to design primer pairs that can uniquely detect murine viruses we identified through our bioinformatics analysis: XMLV *ENV* primer pairs for *ENV* gene of murine type C retrovirus: forward 5′-CACCCCCACCGCTCTCAAAG-3′, reverse 5′-GTTG TACCGAGGCTCCTGCC-3′, product size, 193 bp; XMRV *GAG* primer pairs target the *GAG* of Xenotropic MuLV-related virus: forward 5′-TCCGCCGAATGGCCAA CTTT-3′, reverse 5′-GCAGATCGGGACGGAGGTTG-3′, product size, 263 bp; MLV *GPP* primer pairs for *GPP* (gag-pro-pol) of MLV: forward 5′-GCCAGACTGGGGAT CAAGCC-3′, reverse 5′-TGGTGGGGTGGAGTCTCAGG -3′, product size, 283 bp (Supplementary Table 8).

Each PCR reaction mix contained 12.5 µl of Echno PCR Master Mix (2X), 1 µl of Primer mix (10 µM), 2 µl of 500 ng PDX template DNA, and 9.5 µl of ddH$_2$O. The primer pairs (human *GAPDH*) for the human *GAPDH* gene were used as controls to test the reliability of the system. The reaction with no template was used as a negative control. PCR amplification reactions were performed as follows: 95 ℃ (3 min), then 40 cycles at 94 ℃ (30 s)/60 ℃ (45 s)/72 ℃ (50 s), and a final extension at 72 ℃ for 5 min 10 µl of the PCR product was separated by electrophoresis on a 1.8% agarose gel that contained 3 µg/ml ethidium bromide and visualized via FluorChem M imager. PCR amplification was performed twice for each experiment.

**Statistical analysis.** We collected the data used in this project from 184 different experiments. Since many of these datasets lack a normal distribution, all comparisons of virus loads were supported by the Wilcoxon-rank test to be consistent. Data were analyzed using Wilcoxon-rank tests and Pearson correlation tests in the R language package[58]. Box plots were plotted by BoxPlotR[59]. Raw reads mapped to the housekeeping gene actin beta (*ACTB*) were analyzed using HISAT2, Samtools, and HTSeq[57,60,61].

**KEGG pathway analysis.** Differential expression of genes in groups with high and low murine virus contamination was analyzed using HISAT2, Samtools, HTSeq, and EdgeR[57,60–63]. The KEGG enrichment of the top 200 differentially expressed genes was analyzed with the online tools in WebGestalt[64]. The KEGG dot plot was plotted with an R scripts sp_enrichmentPlot.sh[65]. Gene ontology annotation of genes was obtained from gene data in GeneCards[66].

**Phylogenetic analysis.** The *GAG* and *ENV* genes from top murine virus categories (Fig. 2a, b) were searched against the human nr database with a bitscore >200 to identify their homologs. Gag and Env genes from murine viruses and human endogenous retroviruses were aligned using MUSCLE (MUltiple Sequence Comparison by Log-Expectation) with default parameters[67]. The phylogenetic tree was constructed using PhyML in SMS with the Bayesian Information Criterion. The best model (Flu +G+F) was applied[68].

**Reporting summary.** Further information on research design is available in the Nature Research Reporting Summary linked to this article.

## Code availability

All the data used in this work were obtained from Sequence Reads Archive[46]. These datasets are listed in Supplementary Table 2 with their unique accession numbers, and can be freely downloaded using the accession numbers. The remaining data are available within the Article, Supplementary Information or available from the authors upon request. Source data are available as a Source Data file.

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

## Acknowledgements
The authors used computing resources from the Texas Advanced Computing Center at The University of Texas at Austin, the Hopper high performance clusters at Auburn University, and the Data Science and Informatics Core for Cancer Research at the School of Biomedical Informatics, University of Texas Health Science Center at Houston. We would like to thank Drs. Jeff Chang (UTH), Helen Piwnica-Worms (MD Anderson), Gloria Echeverria (Baylor College of Medicine), David Shih (MD Anderson), Hong Jiang (MD Anderson) and Bin Zhang (Baylor College of Medicine) for their valuable help and insight. We also would like to thank Dr. Michael Lewis, Dr. Anadulce Hernandez-Herrera, Mrs. Gabriela Romero, and Mrs. Lacey Dobrolecki from Baylor College of Medicine PDX-AIVM Core (Supported by CPRIT RP170691 and NIH/NCI CA125123) for providing PDX DNA samples. This work is partly supported by the Cancer Prevention and Research Institute of Texas through grant RP170668 (Zheng), PR150551&PR190561 (An), the National Institutes of Health (NIH) through grants 1UL1TR003167 and R01AG066749 (Zheng) and the Welch Foundation AU-0042-20030616 (An). We thank Dr. Georgina T. Salazar for editing the manuscript.

## Author contributions
Z.A., Z.Y., and W.J.Z. conceived the idea; Z.Y. performed analysis; X.F. performed PCR experiments; N.Z., J.J.Z., J.W., H.X., and T.F. provided insightful guidance and suggestions for the project; Z.Y. drafted the manuscript; all authors participated in manuscript revisions; and Z.A. and W.J.Z. oversaw the whole project and finalized the manuscript.

## Competing interests
The authors declare no competing interests.
