## [Peer Review File · Nature Communications]

Reviewers' comments:

Reviewer #1 (Remarks to the Author): Expertise in PDX models

This is an intriguing manuscript in which the authors describe what appears to be a significant potential limitation of PDXs: infection of PDXs with murine viruses. They do an interesting analysis of the "dark" sequencing data that is often ignored in RNAseq experiments and find that a significant portion of these reads correspond to murine viruses. They suggest that these viruses integrate into PDX genomes and affect gene expression and that this may help explain the lack of success translating results from PDX studies into human clinical trials.

Major concerns:

1. A major analysis performed compares PDX from control cancer cells. However, it is very difficult to determine what these control cells are (e.g. line 90) – are the tumors cultured in a xenograft free environment established cell lines? Tumor explants directly from patients? Tumor explants from PDXs? The methods and source of these samples must be made crystal clear.
2. A major concern of all data except the single cell data is that this is bulk sequencing and thus can include both murine cell and cancer cell sequences. This needs to be very clearly laid out.
3. Figure 3 – the authors describe a significant difference in murine virome reads after in vitro culture of pancreatic cancer samples: how long were the cells cultured? Are these pure cancer cell cultures or do they still contain mouse stroma – it can take months to eliminate all fibroblasts from culture. If this experiment was not performed by the authors, then they should provide the proportion of total reads that map to murine genes as a surrogate to understand whether the remaining reads are due to virus infection of murine cells or virus infection of human cells.
4. Likewise – figure 3 – the significant difference between single cell and PDX model reads suggests the high levels are due to contamination with murine cells in the bulk sequencing that is eliminated in the single cell data perhaps by eliminating cells of mouse origin (was this done?)?
5. Lines 176-189 – this is perhaps the most interesting part of this manuscript, but the authors do not seem to provide any data for these "results". Was a figure left out?
6. Discussion – the authors need to be more clear in their discussion of downregulation of immune genes in lymphocytes due to viral infection – PDXs are established in immunocompromised animals. The fact that results from humanized PDXs don't perfectly translate to the clinic is almost certainly multifactorial and not due solely to viral infection of lymphocytes or tumors.
7. How did you define high vs low murine virus contamination? How does this correlate with murine cell reads (see comment 3)?

Minor points:

1. Line 58 – RNA-seq from PDXs also includes a significant proportion of reads originating from murine stroma that must be differentiated from human reads and there is often high homology between the species.
2. Line 158 – the authors refer in fig 5 to cells cultured in a xenograft free environment is this the cell culture with PDX? Please try to use the same nomenclature to refer to the same cells throughout
3. The authors use a review (ref 29) to point to durvalumab failing in bladder cancer but I don't believe the data is that clear and that is certainly not the optimal reference to use.

Nitpicky:

Evidence can be both singular and plural – no need to use evidences

Bevacizumab spelled incorrectly

I hope these comments help improve the usefulness and quality of your manuscript.

R. Kimple

Reviewer #2 (Remarks to the Author): Expertise in PDX models

Nature Comm 19-40495T

This article by Yuan et al investigates the potential contributions of contaminating murine viruses on human PDX collections by analyzing public sequencing data from a large collection of PDX models from different types of cancers. The topic is a highly relevant, important, and an interesting one with potential broad implications and would be of interest to a wide readership. The authors have several good approaches and ideas but the work is unfortunately preliminary and much more specific and detailed analysis is needed to support their conclusions. The novelty is somewhat modest without this type of more in depth analysis and useful analysis since the overall concept that viruses are present in PDX and could cause problems is already documented in a general manner within the literature as they point out. The value here would be for them to use their larger dataset in a careful manner to make some conclusions. Specific suggestions for improvement are as follows:

The authors are too vague and non-specific in their approach to the prior literature and background understanding of virology. Virus transduction and strain/species/tissue tropisms are complex but are well understood and they would benefit from working more closely with a veterinary or human virologist to aid in the conclusions and writing.

The term they use "dark matter" seems not as appropriate since the sequencing data they are referring to are well established to be viral in origin and not of unknown origin. Also another term that seems confusing is "mouse originated".

It would be important in the first section of results and Figure 1 to describe in more detail the exact nature of the models used. Supp 1 contains data that needs to be in the main figures or text somewhere so that readers can find it more accessible as to tumor type and origins. Examples that are critical is that PDX is used as a term for ATCC type older cell lines implanted into mice, primagrafts direct from patients, or from say more recent patient-derived cell lines which are implanted into mice in short time frame. Each of these could affect their results differently and its not clear in the main text what was analyzed.

A critical point in this paper is that they need to better determine and quantify the proportion of human tumor specific reads, mouse host normal tissue specific reads, human virus reads, murine virus reads, shared human/mouse reads (some reads are too homologous to assign to one or the other), and unassigned reads. The fraction for each model for these values could be calculated and provided. This is not discussed at all right now and its hard to interpret their results without knowing how well this has been done.

To this point there is known to be significant mouse host cell contamination at highly varied levels across PDX cancer types (e.g. melanoma may have more mouse normal contamination than say lung cancers). It would be important to first control for the normal cell contribution to the viral content before concluding much about the disproportionate levels in PDX.

The strains of mice used as host is critical to track. Inbred strains can have endogenous retrovirus contributions that are different than another strain and would start to affect I believe their results significantly if each dataset used different hosts. Also they use NSG as host control but do not state whether all the animals used are NSG for the PDX.

The other thing that is not provided in clear fashion is details as to how their "total virome" is calculated. It seems to be the mouse virus reads+human virus reads. Then they are dividing the mouse virus reads by this mouse+human virus number.

The Figure 1a is confusing in that it seems to show no significant difference in viral reads between immunocompromised NSG (n=3) and wild type mice (n=5). This seems counterintuitive and in one portion of the text they state that immunocompromised mice have a known higher virus burden. Please explain or cite literature to provide an explanation.

Controls of NSG and wild type mice seem underpowered and would be helpful to have much higher numbers here since the other PDX and human tumors are being analyzed by the hundreds. Maybe the mice selected are inappropriately low rates of virus infection.

The authors need to state in Fig 1 more clearly that the "control" for the cancers is primary human tumor tissue samples I believe.

The finding that murine viral loads could be extremely high in PDX even compared to compromised hosts at baseline is interesting but it seems not entirely clear what hypothesis they have for how this could be occurring?

Some PDX evolve to be replaced by EBV-transformed human lymphocytes which forms tumor. The authors should state whether they see this in the PDX analyzed and to what degree it is present but not noted in the original manuscripts. This would be a nice addition. Also CMV role in human tumors is hotly debated and could be helpful to delineate whether they see it here.

Reference to conclusions from large scale genomics analysis of pathogens in cancer samples would be helpful. What types of viruses are present in the human tumors as context.

Fig 1C does not seem to add much.

Fig 2a the pie chart has no legend or labels as to the proportion of each virus contribution?

A table listing the murine and human viruses detected and their known species and tissue tropism would help the non-virologist readers to understand the possibilities here better. Its important to know which murine viruses could infect human cells naturally and distinguish those from murine viruses which cannot infect human cells.

Many of the statements in the manuscript do not use statistics to support the claim.

Reviewer #3 (Remarks to the Author): Expertise in viral metagenomics

The authors have used metagenomics to identify viruses present in patient-derived xenografts. They report "pervasive viral infection" of a majority of the xenografts. Furthermore, they report that the viruses "lowered the expression of immune-related genes." The viruses also altered expression of metabolism and genes that been associated with tumorigenesis.

This study addresses an important issue, namely, are the biological properties of patient-derived xenografts altered by infection with viruses coming from the mouse host. As pointed out by the authors, important parameters, such as drug metabolism, could be altered by infection thus compromising the use of xenografts in a number of important studies.

While this manuscript focuses on an issue of great interest, a number of critical issues need to be addressed.

Comments:

1. Other than murine retroviruses it is not clear what the authors found when searching NGS "dark matter." Sequence reads may not align to the human genome for a number of reasons. Sequencing errors or polymorphisms distinct from human reference genomes are foremost. Bacteria, viruses, and other organisms may also be present. What percentage of reads in each NGS study constituted "dark matter?" How many of these did not align to anything in the databases? Did any align to viruses other than murine retroviruses?

2. Figure 1a. The authors must explicitly describe how the "Relative proportion of murine

originated virus in the total virome" is calculated. This sounds as if it should be the number of reads aligning to murine retroviruses divided by the total number of viral reads. However, the virome of these samples is not described. Were other viruses detected?

3. Since retroviruses have RNA genomes, NGS will detect both virion genomic RNA as well as viral mRNAs. The viral sequences from each sample should be assembled to determine if the authors are detecting genomic RNA or mRNA.

4. The authors detected chimeric reads containing both viral and human sequences (Figure 4). Chimeric reads are often indicative of viral integration. However, chimeric reads can also be generated as an artifact of NGS. The key to distinguishing between these two possibilities is the read coverage of the host-virus junction. For true integration events, multiple chimeric reads should identify the same precise junction. The authors should report the coverage of each junction.

5. Each viral integration results in two host-virus junctions, one at each end of the insertion. By pairing matched junctions, it is often possible to determine where the breakpoints are in the viral genome, and what portions of the viral genome are integrated. Retroviral integrations occur via the LTRs. Do the authors detect full viral integrations? Where do the host-virus junctions map on the viral genome? Do they make biological sense?

6. The authors have identified changes in cellular gene expression that correlate with "viral load" in one sample. How do they distinguish between the changes in gene expression being due to infection (the author's interpretation), from the possibility that cells with a specific gene expression pattern are more susceptible to infection?

7. Legend for Figure 6. This figure describes gene expression profiles in lung cancer xenografts with high and low viral loads. How does the last sentence referencing bladder cancers fit in?

8. In the Introduction (starting line 60) the authors state that the "dark matter" of NGS data, that is the sequences that do not align to the human genome, are discarded. This may be true for xenograft studies. However, there are many published papers and much software, directed towards analysis of the "dark matter" in NGS data and these have proved to be a rich source for viral discovery, and for assessing the effects of viral infection on host gene expression patterns. These works should be acknowledged and cited.

**“Dark Matter” of Sequencing Data Reveals Patient-Derived Xenografts are Extensively**
**Compromised by Pervasive Viral Infection (NCOMMS-19-40495-T)**

We appreciate the reviewers’ thorough review, insightful comments, and helpful suggestions for
our manuscript. We have responded to all suggestions and believe our revised manuscript is
significantly improved. Detailed responses to the reviewers’ comments are listed below; we
marked the revisions in red in the manuscript.

**Reviewer #1 (Remarks to the Author): Expertise in PDX models**

Major concerns:

*1. A major analysis performed compares PDX from control cancer cells. However, it is very*
*difficult to determine what these control cells are (e.g. line 90) – are the tumors cultured in a*
*xenograft free environment established cell lines? Tumor explants directly from patients? Tumor*
*explants from PDXs? The methods and source of these samples must be made crystal clear.*

**Response:** We have now described these sources in the first sentence of “Data” in the
Methods section (line 307-314), and summarized the information in a new Supplementary Table
1a. The origins and sources of the samples are given in Table 1 and the new Supplementary
Table 1b.

*2. A major concern of all data except the single cell data is that this is bulk sequencing and thus*
*can include both murine cell and cancer cell sequences. This needs to be very clearly laid out.*

**Response:** We used sequencing data from mouse controls and mapped them first to the
human genome. Unmapped reads were then mapped to the mouse genome. We calculated the
ratio between the number of reads mapped to mouse and human genomes and used this ratio to
infer the total number of mouse reads present in the unmapped reads of PDX tumors, PDX cell
lines, and other analyses in this study. We added these details in the 4th-6th paragraphs of the
section “Presence of murine viruses in PDX models” in Results (line 122-146), and in
Supplementary Table 7.

We then analyzed the correlation between the percent of mouse-specific reads in the
unmapped reads and the percent of murine viral reads in the virome for each sample. Our results,
shown in Figure 1d, indicate that they have no significant correlation (Pearson correlation test:

0.03, p-value = 0.6847). Furthermore, we observed that murine viral loads in most, if not all
 PDX tumors are much higher than those in even wild-type and NSG mice (Figure 1a). We
 concluded that the murine viral reads in PDX tumor sequencing data are mainly from PDX
 tumors, and that levels of viral load are due to viral replication in PDX tumors instead of mouse
 stroma contamination (see second to the last paragraph, “Presence of murine viruses in PDX
 models”, in Results, line 147-155).

 *3. Figure 3 – the authors describe a significant difference in murine virome reads after in vitro*
 *culture of pancreatic cancer samples: how long were the cells cultured? Are these pure cancer*
 *cell cultures or do they still contain mouse stroma – it can take months to eliminate all*
 *fibroblasts from culture. If this experiment was not performed by the authors, then they should*
 *provide the proportion of total reads that map to murine genes as a surrogate to understand*
 *whether the remaining reads are due to virus infection of murine cells or virus infection of*
 *human cells.*

**Response:** Unfortunately, the dataset we used lacks this information. Thus, we updated our
 analysis with a new batch of released data, which contains details regarding numbers of passages
 for cell cultures. As shown in the Figure 2a and the table below, reads that can be mapped to the
 mouse genome are at an extremely low but constant level from 2-20 generations in these cell
 cultures. Furthermore, the level of mouse reads does not correlate with the level of murine virus
 reads detected (Figure 1d).

Run Accession	Passages	Human Unmapped Percentage	Mouse mapped Percentage
SRR5114302	2to3	0.013	3.441E-05
SRR5114303	2to3	0.019	2.827E-05
SRR5114304	2to3	0.015	3.065E-05
SRR5114305	2to3	0.019	2.590E-05
SRR5114306	2to3	0.018	2.644E-05
SRR5114307	2to3	0.013	3.240E-05
SRR5114308	2to3	0.015	2.531E-05

SRR5114309	2to3	0.013	2.928E-05
SRR5114310	2to3	0.014	2.985E-05
SRR5114311	2to3	0.018	3.075E-05
SRR5114312	2to3	0.018	2.798E-05
SRR5114313	2to3	0.020	2.596E-05
SRR5114316	2to3	0.019	2.315E-05
SRR5114317	2to3	0.023	3.401E-05
SRR5114318	2to3	0.018	2.034E-05
SRR5114314	11to20	0.020	2.809E-05
SRR5114315	11to20	0.017	2.247E-05
SRR5114320	11to20	0.011	4.903E-05

*4. Likewise – Figure 3 – the significant difference between single cell and PDX model reads*
*suggests the high levels are due to contamination with murine cells in the bulk sequencing that is*
*eliminated in the single cell data perhaps by eliminating cells of mouse origin (was this done)?*

**Response:** The data indeed suggest that bulk sequencing contains murine cells, since we
found mouse-specific sequences from PDX tumors (Supplementary Table 5). However, we also
found no direct correlation between the amount of mouse reads and murine viral load (Figure
1d). There could be multiple reasons for lower viral reads in single cell sequencing compared to
PDX tumors. The two most likely reasons are: 1) In many cell cultures, drugs that prevent
microbes (mycoplasma and viruses) are added to prevent contamination, which could inhibit the
normal growth of viruses; or 2) low RNA signals from a single cell (Supplementary Table 8).

Sequencing reads generated from a single cell are much lower than in bulk sequencing, as
very low amounts of RNA are recovered from each cell (Supplementary Table 8). On the other
hand, viral RNA molecules make up a small portion of the total RNAs from a host cell
(Supplementary Table 5). Thus, viral RNAs detected in single sequencing will be
disproportionally low due to low RNA levels and reduced sensitivity of single-cell sequencing.
Nevertheless, we observed a significant amount of murine viral reads in each single cell derived
from PDX tumors but not from primary tumors (Figure 2b), which supports our hypothesis that
murine viruses are present in PDX tumor cells.

*5. Lines 176-189 – this is perhaps the most interesting part of this manuscript, but the authors do*
*not seem to provide any data for these “results”. Was a Figure left out?*

**Response:** We have added Figure 5d to show that in human tumor cells infected by oncolytic
viruses, some genes showed expression change similar to those in high murine virus load PDX
tumors. We also modified this section of the manuscript to eliminate speculation (line 239-241).

*6. Discussion – the authors need to be clearer in their discussion of downregulation of immune*
*genes in lymphocytes due to viral infection – PDXs are established in immunocompromised*
*animals. The fact that results from humanized PDXs don’t perfectly translate to the clinic is*
*almost certainly multifactorial and not due solely to viral infection of lymphocytes or tumors.*

**Response:** We think our results are important in two ways. First, for conventional small
molecule drug development, PDXs are transplanted to immunocompromised animals but contain
many types of immune cells. Therefore, our discussion about the expression of immune genes is
relevant in this context as they may influence the drug responses of PDX tumor. In addition,
more and more drug discovery projects are focused on immunotherapy. Our observation of
immune gene expression would have potential impact on this type of drug discovery. We
modified the second paragraph of the Discussion and added a sentence to the end of the
paragraph to clarify these points (line 273-275).

*7. How did you define high vs low murine virus contamination? How does this correlate with*
*murine cell reads (see comment 3)?*

**Response:** We quantified the murine virus reads in each sample and ranked the samples by
the proportion of murine virus reads compared to the whole virome of the sample (see “Reads
Mapping and Analysis” in the Methods section, line 331-334). The murine cell reads were
analyzed as described in our response to Comment #2. As discussed above in Comment #3, we
saw no significant correlation between the murine virus reads and the murine cell reads,
indicating that murine virus reads are not contaminations.

*Minor points:*

*1. Line 58 – RNA-seq from PDXs also includes a significant proportion of reads originating*
*from murine stroma that must be differentiated from human reads and there is often high*
*homology between the species.*

**Response:** We have modified this sentence (Second sentence of the third paragraph of
Introduction, line 57-60).

*2. Line 158 – the authors refer in fig 5 to cells cultured in a xenograft free environment is this*
*the cell culture with PDX? Please try to use the same nomenclature to refer to the same cells*
*throughout.*

**Response:** Due to revisions in this manuscript, Figure 5 is now Figure 4. Those samples
came from primary cell cultures. We have also added detailed nomenclature in Supplementary
Table 1a and used it throughout the manuscript.

*3. The authors use a review (ref 29) to point to durvalumab failing in bladder cancer but I don't*
*believe the data is that clear and that is certainly not the optimal reference to use.*

**Response:** There are many reports of clinical trials for durvalumab, with various results. For
example, one paper [Stewart et al, Cancer Immunol Res. 2015 3:1052-62] suggests that
durvalumab (MEDI4736) has antitumor activity in pancreatic tumor cell lines in mouse
xenografts (page 1057). But a more recent paper states that durvalumab failed in patients with
pancreatic cancer in a Phase II clinical trial [O'Reilly et al, JAMA Oncol. 2019 Jul 18. doi:
10.1001/jamaoncol.2019.1588.]. We have updated the reference in the manuscript for these
clinical trials (Ref 35, 36).

Nitpicky:

*1. Evidence can be both singular and plural – no need to use evidences. Corrected.*

*2. Bevacizumab spelled incorrectly. Corrected.*

**Reviewer #2 (Remarks to the Author): Expertise in PDX models**

*1. The authors are too vague and non-specific in their approach to the prior literature and*
*background understanding of virology. Virus transduction and strain/species/tissue tropisms are*
*complex but are well understood and they would benefit from working more closely with a*
*veterinary or human virologist to aid in the conclusions and writing.*

**Response:** We modified and acknowledged the previous work at the beginning of the fourth
paragraph in the Introduction section (line 70-72). We also added Supplementary Table 3, which
lists the 25 murine viruses we identified and their potential hosts. Finally, we discussed our work
with Dr. Hong Jiang, a virologist at MD Anderson who is developing oncolytic viral solutions
for tumor therapy. Her suggestions were incorporated into our manuscript, and we note her
contribution in the Acknowledgements section.

*2. The term they use “dark matter” seems not as appropriate since the sequencing data they are*
*referring to are well established to be viral in origin and not of unknown origin. Also another*
*term that seems confusing is “mouse originated”.*

**Response:** We use the term “dark matter” here because our data are from experiments where
the focus was on the human genome. Typically, after investigators map most of the reads to the
human reference genome, they discard unmapped reads without further analysis. In that sense,
these unmapped reads are the “dark matter” of genome sequencing data. Our work suggests that
these unmapped reads still contain meaningful information. We think that by calling attention to
unmapped reads, we can raise awareness of their potential importance.

By using the term “mouse originated”, we wanted to emphasize that these viruses are from
mice via xenografts, not from patients. But we agree that this could be confusing, so we have
removed this term from the manuscript.

*3. It would be important in the first section of results and Figure 1 to describe in more detail the*
*exact nature of the models used. Supp 1 contains data that needs to be in the main Figures or*
*text somewhere so that readers can find it more accessible as to tumor type and origins.*
*Examples that are critical is that PDX is used as a term for ATCC type older cell lines implanted*
*into mice, primagrafts direct from patients, or from say more recent patient-derived cell lines*

*which are implanted into mice in short time frame. Each of these could affect their results*
*differently and its not clear in the main text what was analyzed.*

**Response:** We have added Supplementary Table 1a to list our nomenclature for the samples
and now use these terms throughout the manuscript. In addition, we moved content from original
Supplementary Table 1 to Table 1 to clarify the source of the samples. We also added detailed
descriptions of these samples and nomenclature at the beginning of “Data”, Methods section
(line 307-314).

*4. A critical point in this paper is that they need to better determine and quantify the proportion*
*of human tumor specific reads, mouse host normal tissue specific reads, human virus reads,*
*murine virus reads, shared human/mouse reads (some reads are too homologous to assign to one*
*or the other), and unassigned reads. The fraction for each model for these values could be*
*calculated and provided. This is not discussed at all right now and its hard to interpret their*
*results without knowing how well this has been done.*

**Response:** We performed the suggested analysis; results are included in Supplementary
Table 5. We also revised “Reads Mapping and Analysis” in the Methods section (line 327-341)
and the 4th-6th paragraphs in “Presence of murine viruses in PDX models”, Results section (line
122-146).

*5. To this point there is known to be significant mouse host cell contamination at highly varied*
*levels across PDX cancer types (e.g. melanoma may have more mouse normal contamination*
*than say lung cancers). It would be important to first control for the normal cell contribution to*
*the viral content before concluding much about the disproportionate levels in PDX.*

**Response:** Please see our response to comment #4. We found that: mouse-specific reads are
present in the unmapped reads (Supplementary Table 5); viral load in almost all types of PDX
tumors is much higher than in wild-type and NSG mice (Figure 1a); and murine viral load and
mouse-specific reads are not correlated (see Supplementary Table 5 and Figure 1d).

*6. The strains of mice used as host is critical to track. Inbred strains can have endogenous*
*retrovirus contributions that are different than another strain and would start to affect I believe*

*their results significantly if each dataset used different hosts. Also they use NSG as host control*
*but do not state whether all the animals used are NSG for the PDX.*

**Response:** The PDX is based on the immune deficiency mouse model used by Okada *et al*
(2019; see Okada¹ below). We listed all sources for the PDX models in Supplementary Table 1b
and provided the mouse host strain information if known. We acknowledge that the original data
do not list information on strains. However, we found that murine viruses are still the dominant
types in the total virome regardless of the mouse strain (Supplementary Table 5).

*7. The other thing that is not provided in clear fashion is details as to how their “total virome” is*
*calculated. It seems to be the mouse virus reads+human virus reads. Then they are dividing the*
*mouse virus reads by this mouse+human virus number.*

**Response:** Yes, that is correct. We added this definition in the first paragraph of “Presence of
Murine Viruses in PDX Models”, Results section (line 93-94).

*8. The Figure 1a is confusing in that it seems to show no significant difference in viral reads*
*between immunocompromised NSG (n=3) and wild type mice (n=5). This seems counterintuitive*
*and in one portion of the text they state that immunocompromised mice have a known higher*
*virus burden. Please explain or cite literature to provide an explanation. Controls of NSG and*
*wild type mice seem underpowered and would be helpful to have much higher numbers here*
*since the other PDX and human tumors are being analyzed by the hundreds. Maybe the mice*
*selected are inappropriately low rates of virus infection. May be, the sample sizes are limited.*

**Response:** We carefully reviewed our analysis pipeline and rechecked the content of virus
reads and their mapping. We found that some reads from wild-type mouse samples mapped to a
few artificial sequences incorporated into the ReadScan database as viral genome sequences.
These artificial sequences actually come from patented sequences. They are not full viral genome
sequences, but contain synthetic construct or vector sequences, and only included a small
proportion of the virus genome. We fixed the ReadScan database issues, identified more wild-
type and NSG samples, reran all our analyses using the updated database, and updated the results
throughout the manuscript. The new results show that NSG mice have a higher murine viral
proportion (p-value = 1.72E-04) than wild-type mice (Figure 1a). Our inspection shows that all
the other analysis results are not impacted by this issue.

9. The authors needs to state in Fig 1 more clearly that the “control” for the cancers is primary human tumor tissue samples I believe.

Response: We have added Table 1 to clearly indicate if the control is from corresponding primary cancer tissue or cell lines. We also updated this information in the Figure legend.

10. The finding that murine viral loads could be extremely high in PDX even compared to compromised hosts at baseline is interesting but it seems not entirely clear what hypothesis they have for how this could be occurring?

Response: Tumor cells are prone to viral infection and virus growth due to their lack of robust anti-viral mechanisms². We surmise that as a result, PDX tumors become an even greater growth hub than the immunocompromised host. If so, most murine viral reads in PDX tumors are likely from viral infection of the PDX tumor, not contamination of mouse stroma, since the murine virus in the host cannot reach the same level as in the PDX tumor. We discuss this point in the last sentence of the second to the last paragraph, “Presence of murine virus in PDX models”, Results section (line 151-155).

11. Some PDX evolve to be replaced by EBV-transformed human lymphocytes which forms tumor. The authors should state whether they see this in the PDX analyzed and to what degree it is present but not noted in the original manuscripts. This would be a nice addition.

Response: EBV transformation has indeed been observed in several cancer types, such as in the PCAWG project³ and PDX samples^{4,5}. The presence of EBV in PDX also can be associated with formation of human lymphocytic tumors in PDX⁶. We did observe the existence of EBV in some studies, such as ERR1084820 and ERR1084816. In those samples, EBV reads were identified covering the whole genomic region (see Supplementary Figure 5). We added a new paragraph at the end of “Presence of murine viruses in PDX models” in the Results section (line 156-162) to discuss this topic and HCMV (see below).

12. Also CMV role in human tumors is hotly debated and could be helpful to delineate whether they see it here.

**Response:** We did not observe HCMV contigs in our study. We added a new paragraph at
the end of “Presence of murine viruses in PDX models” in the Results section on this topic. The
HCMV is discussed in the last sentence (line 161-162).

*13. Reference to conclusions from large scale genomics analysis of pathogens in cancer samples*
*would be helpful. What types of viruses are present in the human tumors as context.*

**Response:** We have added new references to studies targeting the content of the cancer
virosphere and their association with human cancer^{3,7,8} at the beginning of the fourth paragraph
of the Introduction section (line 70-72). As discussed in our response to comment #11, we
observed some viruses, such as HPV type 16 or EBV4, in our analysis. The full list of viruses
we discovered are shown in the Supplementary Table 4.

*14. Fig 1C does not seem to add much.*

**Response:** Figure 1C was another way to show that each PDX tumor type is different from
the others, while all the controls behave the same. However, we have moved these data to
Supplementary Figure 1.

*15. Fig 2a the pie chart has no legend or labels as to the proportion of each virus contribution?*

**Response:** The original Figure 2a was merged into Figure 1b. We added the labels in the
Figure.

*16. A table listing the murine and human viruses detected and their known species and tissue*
*tropism would help the non-virologist readers to understand the possibilities here better. Its*
*important to know which murine viruses could infect human cells naturally and distinguish those*
*from murine viruses which cannot infect human cells.*

**Response:** We identified 25 viruses of murine origin (see Supplementary Table 3).

*17. Many of the statements in the manuscript do not use statistics to support the claim.*

**Response:** Since we collected these data from hundreds of research groups, not all the data
are normally distributed. Therefore, all comparisons of data are supported by the Wilcoxon-rank
test and correlations were conducted using the Pearson correlation test. We have added new text

about this consideration as described in the Methods section (line344-346). We also tightened
our manuscript to ensure that statements are supported by statistical analysis.

**Reviewer #3 (Remarks to the Author): Expertise in viral metagenomics**

*1. Other than murine retroviruses it is not clear what the authors found when searching NGS*
*“dark matter.” Sequence reads may not align to the human genome for a number of reasons.*
*Sequencing errors or polymorphisms distinct from human reference genomes are foremost.*
*Bacteria, viruses, and other organisms may also be present. What percentage of reads in each*
*NGS study constituted “dark matter?” How many of these did not align to anything in the*
*databases? Did any align to viruses other than murine retroviruses?*

**Response:** We used the software ReadScan for virus detection. To address the reviewer’s
question, we added two tables: Supplementary Table 4 lists all viruses detected in each sample.
Supplementary Table 5 lists details about reads that mapped to the human, mouse, or virus
genome, and the numbers of reads not mapped to anything (Columns XIII and XIV). In
Supplementary Table 4, we reported observed human and other viruses, such as HPV type 16 or
EBV4. We also observed some phage 174 sequences, consistent with published observations and
believed to derive from sequencing contamination^{3,9}.

*2. Figure 1a. The authors must explicitly describe how the “Relative proportion of murine*
*originated virus in the total virome” is calculated. This sounds as if it should be the number of*
*reads aligning to murine retroviruses divided by the total number of viral reads. However, the*
*virome of these samples is not described. Were other viruses detected?*

**Response:** Please see the end of the second paragraph, “Reads Mapping and Analysis” in the
Methods section (line 331-334). As discussed above, we did observe other viruses, presented in
Supplementary Table 4. We also analyzed the assembled contigs for some of these viruses in
Supplementary Figures 3 and 5.

*3. Since retroviruses have RNA genomes, NGS will detect both virion genomic RNA as well as*
*viral mRNAs. The viral sequences from each sample should be assembled to determine if the*
*authors are detecting genomic RNA or mRNA.*

**Response:** We assembled all unmapped reads into distinct contigs. Reads of the major
murine viruses we observed can be assembled into a complete genome, indicating the existence
of a full viral genome (see Supplementary Figures 3-5). But the depth of reads is uneven across

the viral genome, suggesting that the virus RNA contains both genomic RNA and mRNA
(Supplementary Figures 3-5). We revised the manuscript to discuss this point in the third
paragraph of “Presence of murine virus in PDX models”, Results section (line 113-121). We
could not assemble other viruses with less abundant RNA reads into the full genome. Therefore,
we did not draw any conclusions about these viruses, due to insufficient amounts of viral RNA in
tumor cells.

*4. The authors detected chimeric reads containing both viral and human sequences (Figure 4).
Chimeric reads are often indicative of viral integration. However, chimeric reads can also be
generated as an artifact of NGS. The key to distinguishing between these two possibilities is the
read coverage of the host-virus junction. For true integration events, multiple chimeric reads
should identify the same precise junction. The authors should report the coverage of each
junction.*

**Response:** The original Figure 4 is now Figure 3. We used Virus-Clip¹⁰, software designed
to detect viral integration, to perform the analyses that the reviewer suggests. We analyzed the
viral integration sites in six samples with whole genome assembly (Figure 3, Supplementary
Table 9); coverage of reads for these integration sites ranged from 10 to 242 (Figure 3). Thus,
these chimeric reads are not likely from sequencing artifacts. Details about all integration sites
are provided in Figure 3. We also updated the first paragraph in “Integration of murine virus
DNA in PDX tumor genome”, Results section (line 193-195).

*5. Each viral integration results in two host-virus junctions, one at each end of the insertion. By
pairing matched junctions, it is often possible to determine where the breakpoints are in the viral
genome, and what portions of the viral genome are integrated. Retroviral integrations occur via
the LTRs. Do the authors detect full viral integrations? Where do the host-virus junctions map on
the viral genome? Do they make biological sense?*

**Response:** We did not detect both ends of the full integration in our analysis. Since we are
using RNA-Seq data to identify these integration junctions, the other end of the integration site
may not fall into a region of expression, and thus the corresponding chimeric RNA is undetected.
This is very likely, as the viral genome is long. However, as discussed above, each integration

site is supported by a read depth ranging from 10-242. Therefore, it is unlikely our observation is
an artifact. Integration sites and the corresponding genome coordinates are shown in Figure 3.

We observed some virus-host junctions in LTR regions of virus and human exon sequences
with moderate coverage (e.g. Left Position (virus)—7905, Right (human)—DDB1, coverage
72×). These are highlighted in Supplementary Table 9. In addition, previous research suggests
that subgenomic viral fragments or defective virus can occur in retrovirus-cell DNA
junctions^{11,12}; we observed a similar pattern in our analysis (see Supplementary Table 9).

*6. The authors have identified changes in cellular gene expression that correlate with “viral*
*load” in one sample. How do they distinguish between the changes in gene expression being due*
*to infection (the author’s interpretation), from the possibility that cells with a specific gene*
*expression pattern are more susceptible to infection?*

**Response:** We performed a principal component analysis on all PDX samples in this study
together with the corresponding primary tumor controls using gene expression profiling. All
primary tumors formed a tight cluster, while PDX samples are scattered at significant distances
from each other (Figure 5c). This result indicates that the primary tumors have very similar gene
expression profiles before being transplanted into mice for PDX studies, and thus they are
unlikely to contribute to different levels of virus load in the PDX. It is more likely that variation
of gene expression occurred after primary tumors are transplanted and infected by virus. We
have added text on this point in the second paragraph of “Impact of murine viral infection on
PDX gene expressions”, Results section (228-236).

*7. Legend for Figure 6. This Figure describes gene expression profiles in lung cancer xenografts*
*with high and low viral loads. How does the last sentence referencing bladder cancers fit in?*

**Response:** We consolidated the original Figure 6 into Figure 5. The last sentence the
reviewer refers to is actually from the Figure 7 legend stating that “Transcription profiles of
pectilisib treated bladder cancer PDX samples with highest and lowest viral amounts”. This was
an error and we have corrected it.

*8. In the Introduction (starting line 60) the authors state that the “dark matter” of NGS data,*
*that is the sequences that do not align to the human genome, are discarded. This may be true for*

*xenograft studies. However, there are many published papers and much software, directed*
*towards analysis of the “dark matter” in NGS data and these have proved to be a rich source for*
*viral discovery, and for assessing the effects of viral infection on host gene expression patterns.*
*These works should be acknowledged and cited.*

**Response:** We have acknowledged these works and cited it at the beginning of the fourth
paragraph of the Introduction (line 72). The landscape of the virome and its associations with
human cancer have been characterized from TCGA datasets⁷ and the Pan-Cancer Analysis of
Whole Genomes (PCAWG)³. We also acknowledged this work at the beginning of the same
paragraph (line 70-72).

**References:**

- 1. Okada S, Vaeteewoottacharn K, Kariya R. Application of highly immunocompromised mice
for the establishment of patient-derived xenograft (PDX) models. *Cells* 2019;8:889.
- 2. Howells A, Marelli G, Lemoine NR, Wang Y. Oncolytic Viruses-Interaction of Virus and
Tumor Cells in the Battle to Eliminate Cancer. *Front Oncol* 2017;7:195.
- 3. Zapatka M, Borozan I, Brewer DS, et al. The landscape of viral associations in human
cancers. *Nature Genetics* 2020:1-12.
- 4. Fujii E, Kato A, Chen YJ, Matsubara K, Ohnishi Y, Suzuki M. Characterization of EBV-related
lymphoproliferative lesions arising in donor lymphocytes of transplanted human tumor
tissues in the NOG mouse. *Experimental animals* 2014;63:289-96.
- 5. Chen K, Ahmed S, Adeyi O, Dick JE, Ghanekar A. Human solid tumor xenografts in
immunodeficient mice are vulnerable to lymphomagenesis associated with Epstein-Barr
virus. *PloS one* 2012;7.
- 6. Bondarenko G, Ugolkov A, Rohan S, et al. Patient-derived tumor xenografts are susceptible
to formation of human lymphocytic tumors. *Neoplasia* 2015;17:735-41.
- 7. Tang K-W, Alaei-Mahabadi B, Samuelsson T, Lindh M, Larsson E. The landscape of viral
expression and host gene fusion and adaptation in human cancer. *Nature communications*
2013;4:2513.
- 8. Thompson MP, Kurzrock R. Epstein-Barr virus and cancer. *Clinical Cancer Research*
2004;10:803-21.
- 9. Mukherjee S, Huntemann M, Ivanova N, Kyrpides NC, Pati A. Large-scale contamination of
microbial isolate genomes by Illumina PhiX control. *Standards in genomic sciences*
2015;10:18.
- 10. Ho DW, Sze KM, Ng IO. Virus-Clip: a fast and memory-efficient viral integration site
detection tool at single-base resolution with annotation capability. *Oncotarget*
2015;6:20959.
- 11. Risser R, Horowitz JM, McCubrey J. Endogenous mouse leukemia viruses. *Annual review of*
*genetics* 1983;17:85-121.

- 12. Iwase SC, Miyazato P, Katsuya H, et al. HIV-1 DNA-capture-seq is a useful tool for the
comprehensive characterization of HIV-1 provirus. *Scientific reports* 2019;9:1-12.

Reviewer #1 (Remarks to the Author):

Overall the authors did a very good job responding to reviewer comments. I have 2 other simple questions that I think will help clarify the manuscript and results.

I apologize if I missed this the first time – Figure 4 – what is the difference between Non-PDX facility and PDX facility? Do you mean cells growing in culture after first being in a mouse vs. going direct from patient to culture? Or do you literally mean that cells put into culture in the same building/facility as they perform PDX work have higher murine viral reads even though they are never grown in mice?

My understanding is that you first mapped reads to human and then to murine genome. What if you reverse this – do you see different trends?

Reviewer #2 (Remarks to the Author):

NCOMMS-19-40495-T

The authors have done a number of revisions to the manuscript and it is greatly appreciated that they have responded and addressed several of the comments. The paper is significantly improved. However, there do seem to be still some major concerns about analysis approach (even more so after their discovery of errors in their original analysis) and in some of the new data issues with their conclusions. The figure quality is not improved significantly enough at this point in my opinion. This work is very interesting but requires a great deal of rigor in several specialties. Specific areas still of concern are as follows:

1. The authors seem to have mainly just added supplemental figures to address reviewer concerns but many of these points are now deserving of being reported in main figures. The list of all viruses and their reads etc. is at the core of the data and should be in the main figures more properly.
2. The authors now clearly show in the supp table 3 one of the concerns that I had. It seems that they see great number of reads from viruses that are known not to infect human hosts (AKR, AKT8). They suggest this is due to abnormal tumor cells lacking the ability to fend off the viruses in a general sense with one reference. This is really not likely the case by my understanding and so this seems to really cut under their arguments and suggest that somehow there is a problem with the mapping or analysis approach.
3. Their suggestions about the prevalence of viral genomes altering the biology is important and concerning. Given this the authors still did not add any data from orthogonal sources of experimentation to validate their findings in PDX tissues or cells. This is easily done via Western, electron microscopy, or other approaches to directly identify productive infection. The virus should also be present in the supernatant of the cell cultures as well and could be detected perhaps by infection of another culture. Any approach at all would be helpful to more directly show that this is not an informatics problem or a problem with MLV for instance being the backbone of many engineered PDX and cell line models and detection being caused by intentional infection of the cells for luciferization etc.
4. The main figures still are very difficult to read and not well prepared. The pie chart in Fig 1B lacks simple legends for identification of the percentages etc.
5. Ref 20 is a secondary reference and is not the primary reference proposing infection with CMV.
6. Supp Table 4 seems to indicate polyviral infection in the majority of PDX and normal mice. This seems not to change much in the main players on a qualitative level across tumors types and other aspects raising concerns about the mapping of reads again being a main issue and driver of the reads. It would be expected that you might see more of a difference and fewer polyviral infections than this in human cells.

Reviewer #3 (Remarks to the Author):

The authors have met my concerns.

Reviewer #1 (Remarks to the Author):

1. I apologize if I missed this the first time – Figure 4 – what is the difference between Non-PDX facility and PDX facility? Do you mean cells growing in culture after first being in a mouse vs. going direct from patient to culture? Or do you literally mean that cells put into culture in the same building/facility as they perform PDX work have higher murine viral reads even though they are never grown in mice?

Yes, we mean that cells grown in the same facilities that performed PDX have higher viral loads than those grown in facilities that do not perform PDX, or those directly obtained from patients without going through cell cultures. We have updated the Figure 5 and the last paragraph of “Integration of murine virus DNA in the PDX tumor genome” section in “Results” to clarify this point (lines 221-222). This is in agreement with previous observations for cross-lab contamination of murine leukemia virus validated by qPCR assays targeting the gag, env, or pol regions¹.

2. My understanding is that you first mapped reads to human and then to murine genome. What if you reverse this – do you see different trends?

Although we did not perform the suggested analysis in our last submission, we did take an approach that we think is more relevant. Specifically, we took RNA-seq data sets from 14 mouse samples and mapped the mouse reads to the human reference genome; we then mapped the remaining unmapped reads to the murine genome. Next, we calculated the ratio of the amount of mouse reads mapped to the human genome, and used that ratio to adjust for unmapped reads (to human genome) that mapped to the mouse genome from PDX samples. We believe this adjustment could help resolve the cross-mapping issue noted. We describe our approach in the second half of the fourth paragraph of “Presence of murine virus in PDX models” of the “Results” section (lines 131-144) and in Supplementary Table 7. Based on our results, we believe the number of reads mapped to the mouse genome will be slightly higher if we reverse the order as suggested by the reviewer, but this should not be an issue after adjustment by the ratio we calculated.

Reviewer #2 (Remarks to the Author):

Specific areas still of concern are as follows:

1. The authors seem to have mainly just added supplemental figures to address reviewer concerns but many of these points are now deserving of being reported in main figures. The list of all viruses and their reads etc. is at the core of the data and should be in the main figures more properly.

We added a table (Table 2) to list average read counts for viruses detected in PDX and used non-PDX cancer samples as a control. We also split Figure 1 into two figures (Figures 1 and 2) and added panels in Figure 1 (Figure 1c and 1d) to demonstrate the validity of our methodology. We are limited in the numbers of figures and tables we can include. However, we now believe we have included all key information in the figures and tables.

2. The authors now clearly show in the supp table 3 one of the concerns that I had. It seems that they see great number of reads form viruses that are known not to infect human hosts (AKR, AKT8).

Supplementary Table 2, the Virus-Host Database, and other resources only provide information for viruses and their known, experimentally confirmed hosts. However, this information is not an exclusion/negation list, meaning if a host is not listed for a virus, it is due to the lack of experimental evidence whether the virus can infect that host. The absence of a host for a listed virus does not mean there is experimental evidence showing that the virus cannot infect that host, unless experiments/publication(s) are explicitly listed and a non-host role is clearly stated.

Such a situation is evident for several mouse viruses where initial data showed that these viruses infect mouse cells – but no one tested whether they can infect human cells. As such, originally mouse species were listed as the host, but experimental evidence later emerged showing that these viruses can infect human cells. This situation applies to many types of murine viruses such as polytropic or xenotropic murine leukemia virus, which can infect human cells. Our data-driven investigation provides new evidence that these mouse viruses can infect human cells in a PDX setting as shown in Supplementary Table 3 (as pointed out by the reviewer), which has not been reported before. We have added further details about this point in the description of Supplementary Table 2.

3. They suggest this is due to abnormal tumor cells lacking the ability to fend off the viruses in a general sense with one reference. This is really not likely the case by my understanding and so

this seems to really cut under their arguments and suggest that somehow there is a problem with the mapping or analysis approach.

The reduced ability of tumor cells to fend off infection is well documented. Besides the reference the reviewer mentioned, a recent publication showed that different bacteria are found within various tumor cells². We do agree that other factors could contribute to our observation, such as the scenario listed in response to question #7 below. As PDX tumor cells are submerged in fluid full of murine viruses, they could become infected, especially if they have reduced ability to fend off infection. We have added the second reference mentioned above to the manuscript (line 114).

Our analyses are based on well-tested software used in many publications. In addition, many of our findings are consistent with published results, and our new findings do not contradict previously published work (see new Figures 1c and 1d). Furthermore, our findings are cross-validated by many data sets from different research groups concluding that PDX tumors have much higher levels of murine viruses than non-PDX controls. Since all PDX tumors grow in immune-compromised mice infested with murine viruses, it is not surprising that these viruses can infect compromised tumor cells. Therefore, we believe our findings are solid and unlikely to represent random effects from analytic errors.

4. Their suggestions about the prevalence of viral genomes altering the biology is important and concerning. Given this the authors still did not add any data from orthogonal sources of experimentation to validate their findings in PDX tissues or cells. This is easily done via Western, electron microscopy, or other approaches to directly identify productive infection. The virus should also be present in the supernatant of the cell cultures as well and could be detected perhaps by infection of another culture. Any approach at all would be helpful to more directly show that this is not an informatics problem or a problem with MLV for instance being the backbone of many engineered PDX and cell line models and detection being caused by intentional infection of the cells for luciferization etc.

Two lines of evidence validate our findings:

- 1) Several viral infections we observed have already been experimentally validated by published work (see Figures 1c and 1d), which serve to validate our results.
- 2) Our findings are also validated by many data sets from different research groups for different types of cancers and studies. The all-inclusive nature of next-generation sequencing (meaning the sequencing captures everything expressed in the cell) is another important aspect supporting our findings. In addition, PDX tumors consistently have higher levels of murine viruses compared to non-PDX controls. Practically and statistically, such massively cross-validated results assure us about the validity of our findings.

Given the data-driven nature of our approach, the massive amount of data we have analyzed, the consistent findings across different studies by many different research groups, and the validation of our findings by previously published work, we think it is beyond the scope of this work to pursue further experimental validation. However, we do plan to work with other groups to further pursue this project, as suggested by the reviewer.

5. The main figures still are very difficult to read and not well prepared. The pie chart in Fig 1B lacks simple legends for identification of the percentages etc.

We have revised the figures and fixed the issues raised by the reviewer. We also updated Fig 1b and added the missing information.

6. Ref 20 is a secondary reference and is not the primary reference proposing infection with CMV.

We now include information about the first observation of HCMV in glioblastoma in 2002 and added the primary reference (lines 170-173).

7. Supp Table 4 seems to indicate polyviral infection in the majority of PDX and normal mice.

This seems not to change much in the main players on a qualitative level across tumors types and other aspects raising concerns about the mapping of reads again being a main issue and driver of the reads. It would be expected that you might see more of a difference and fewer polyviral infections than this in human cells.

Based on our results and previously published results, we think the scenario for observing murine virus in PDX tumor is as follows: 1) Tumor cells, due to their genomic instability and other changes, are more prone to infection. 2) These infection-prone cells are then placed in immunocompromised mice which have many types of murine viruses. 3) Tumor cells immersed in such an environment then become infected by these murine viruses. This scenario is confirmed by multiple lines of evidence we observed: no direct correlation between mouse and murine viral reads, viral genome integration into the tumor genome, consistent presence of viral sequences in cultured PDX tumor cells through many passages, and the intracellular viral reads from single cell sequencing. Thus, we believe that viral infections are polyviral by nature and are determined by viruses from the murine environment more than PDX tumor cells. Therefore, there may not be a major difference in terms of viral species across different tumor types. We have added this rationale to the manuscript (lines 119-121).

References

- 1 Zhang, Y.-A. *et al.* Frequent detection of infectious xenotropic murine leukemia virus (XMLV) in human cultures established from mouse xenografts. *Cancer biology & therapy* **12**, 617-628 (2011).
- 2 Nejman, D. *et al.* The human tumor microbiome is composed of tumor type-specific intracellular bacteria. *Science* **368**, 973-980, doi:10.1126/science.aay9189 (2020).

Reviewer #2, expert in PDX models (Remarks to the Author):

The attempts to be responsive to reviewer comments are appreciated and this is still potentially interesting work.

However, I still believe that the study is at a preliminary stage of development with unfortunately no evidence of input from expert collaborators with deep knowledge in murine virology and the specific questions at hand which have been studied in fact by many papers in the literature for many decades. This is not to diminish the expertise of the authors in informatics approach but solely due to the complexity of the topic at hand in the area of virology. Also many aspects of the manuscript are still not generally carefully presented in my opinion in such a manner that data can be properly interpreted by readers despite reviewers suggesting improvements in this area.

Reviewer #3, expert in viral metagenomics (Remarks to the Author):

The authors have addressed the concerns of the previous reviews.